# Cytolysin A is an intracellularly induced and secreted cytotoxin of typhoidal *Salmonella*

Lena Krone, Srujita Mahankali ⓘ & Tobias Geiger ⓘ ✉

Typhoidal *Salmonella enterica* serovars, such as Typhi and Paratyphi A, cause severe systemic infections, thereby posing a significant threat as human-adapted pathogens. This study focuses on cytolysin A (ClyA), a virulence factor essential for bacterial dissemination within the human body. We show that ClyA is exclusively expressed by intracellular *S*. Paratyphi A within the *Salmonella*-containing vacuole (SCV), regulated by the PhoP/Q system and SlyA. ClyA localizes in the bacterial periplasm, suggesting potential secretion. Deletion of TtsA, an essential Type 10 Secretion System component, completely abolishes intracellular ClyA detection and its presence in host cell supernatants. Host cells infected with wild-type *S*. Paratyphi A contain substantial ClyA, with supernatants capable of lysing neighboring cells. Notably, ClyA selectively lyses macrophages and erythrocytes while sparing epithelial cells. These findings identify ClyA as an intracellularly induced cytolysin, dependent on the SCV environment and secreted via a Type 10 Secretion System, with specific cytolytic activity.

*Salmonella enterica* serovars such as Typhi (*S*. Typhi) and Paratyphi A (*S*. Paratyphi A) are highly human-adapted bacterial pathogens causing (para)typhoid fever in humans and higher primates. These typhoidal serovars significantly differ in virulence from closely related non-typhoidal, broad host-range serovars of *Salmonella enterica*, which commonly lead to food poisoning. These "generalist" serovars, including *Salmonella enterica* serovar Typhimurium (*S*. Typhimurium), typically cause short-term infections confined to the gastrointestinal tract in healthy humans. In contrast, typhoidal serovars induce life-threatening systemic infections and, for some individuals, result in lifelong chronic infection[1]. Despite sharing many virulence-related genes with closely related non-typhoidal *Salmonella* serovars, *S*. Typhi and *S*. Paratyphi A must possess specific genes pivotal for increased virulence, systemic infection, and human host specificity. Genomic islands shared exclusively among typhoidal serovars, which cause systemic infections, are promising candidates for these specific genes. Comparative genome sequence analyses have identified several typhoidal-specific genomic islands, among which is *Salmonella* pathogenicity island 18 (SPI-18)[2–4]. SPI-18 includes a 1.7 kb operon with two open reading frames, SPA1306 and SPA1307. SPA1307 encodes a putative invasin called Typhi-associated invasin A (TaiA), while

SPA1306 encodes a cytolysin named cytolysin A (ClyA)[5,6]. The sequence of ClyA shares about 91% identity with hemolysin E (HlyE) of *E. coli*, a pore-forming hemolysin demonstrated to lyse monocytes and macrophages[7]. Previous in vitro studies noted similar pore-forming characteristics for ClyA of *S*. Typhi and *S*. Paratyphi A, including the presence of a cholesterol recognition and consensus motif[5,8–10]. When heterologously expressed in *S*. Typhimurium, ClyA induces atypical deep organ infections in mice, emphasizing its contribution to the systemic infection characteristics of typhoidal *Salmonella* serovars[11]. Studies on *S*. Typhi-infected patients detected significant amounts of ClyA and ClyA-specific antibodies in analyzed blood samples, underscoring the clinical relevance of this typhoidal-specific cytolysin[12,13].

ClyA, like most virulence factors, is subject to complex and stringent regulation in response to various environmental cues such as high osmolarity, low pH, and $Mg^{2+}$ conditions[6,14]. Transcription factors including Fis, SlyA, the stress-induced alternative sigma factor RpoS, and the PhoP/Q two-component regulation system play important roles in this regulation[9,14]. Interestingly, transcriptomic studies have shown induced expression of *ClyA* in intracellular *S*. Typhi[6,15]. Another intracellularly induced and pathogenetically important cytotoxin in typhoidal *Salmonella* is the typhoid toxin[16]. This AB toxin, induces

Max von Pettenkofer-Institute, Chair for Medical Microbiology and Hygiene, Ludwig-Maximilians-Universität München (LMU Munich), Munich, Germany.
✉e-mail: geiger@mvp.lmu.de

cytolethal cell distending phenotypes in intoxicated cells due to the DNase activity of its enzymatically active "A" subunit[17,18]. Once synthesized by intracellular *S*. Typhi, typhoid toxin is secreted from the bacteria into the lumen of the *Salmonella*-containing vacuole (SCV) by a novel protein secretion system named Type 10 Secretion System (T10SS)[19–21]. The pivotal component of this secretion system is TtsA (typhoid toxin secretion protein A), a specialized peptidoglycan hydrolase responsible for actively releasing the toxin through the bacterial cell wall without causing bacterial cell lysis[20,22].

In this study, we demonstrate that ClyA, akin to the typhoid toxin in *S*. Typhi, is exclusively expressed and secreted by intracellular *S*. Paratyphi A within the SCV. Moreover, we confirm that the PhoP/Q two-component regulator and the transcriptional regulator SlyA are crucial for intravacuolar ClyA expression. Post-induction, ClyA localizes within the bacterial periplasm, hinting at potential secretion from intracellular bacteria. Consistent with these findings, cells infected with wild-type *S*. Paratyphi A contain substantial ClyA in the cell cytoplasm and subsequently in the host cell supernatants. Intriguingly, we show that ClyA secretion relies on a Type 10 protein secretion system, first described for typhoid toxin secretion in *S*. Typhi. Upon secretion, ClyA exhibits cell-type-specific cytolytic activities. Produced within infected epithelial cells and subsequently secreted, ClyA selectively lyses macrophages and erythrocytes. Conversely, epithelial cells treated with ClyA-containing supernatants remain insensitive to ClyA-mediated cell lysis.

## Results

### ClyA of *S*. Paratyphi A represents a cytolysin induced by intracellular bacteria residing within the *Salmonella*-containing vacuole

The initial indications that ClyA represents a cytolysin induced by intracellular conditions were derived from studies on the regulation of its expression. Primarily, the regulatory impact of the PhoP/PhoQ two-component system on ClyA expression was observed[14]. This system, as demonstrated by numerous studies, plays a crucial role in the intracellular stress responses of *Salmonella* during host cell infection[23]. Consistent with these findings, previous transcriptome and qRT-PCR analyses have suggested that *clyA* is induced within host cells[6,15,24]. In our study, to facilitate the detection of intracellular ClyA at the protein level, ClyA was chromosomally tagged with an N-terminal 3xFLAG epitope tag. Western blot analyses revealed ClyA expression in wild-type bacteria grown in a chemically defined low $Mg^{2+}$ medium known to induce PhoP/Q-regulated proteins (Fig. 1a)[20]. The impact of the PhoP/Q system and the transcriptional regulator SlyA on ClyA expression was demonstrated by mutants in the corresponding *phoP* or *slyA* genes, respectively. Interestingly, the growth of bacteria in LB rich medium did not result in detectable ClyA expression, including during extended growth studies up to 24 h to stationary phase (supplementary Fig. 1a). To investigate whether ClyA is induced within infected host cells, HeLa epithelial cells and isolated peripheral blood mononuclear cells (PBMCs) were infected with wild-type *S*. Paratyphi A expressing chromosomally encoded ClyA:3xFLAG. After 24 h of incubation for HeLa cells and 2 h for PBMCs, intracellular bacteria were harvested, and ClyA expression was analyzed by Western blot analyses. The ClyA expression pattern indicated a strong induction of ClyA within HeLa epithelial cells and primary immune cells (Fig. 1b, c). Similar to the low $Mg^{2+}$ medium, the expression depended entirely on the presence of a functional PhoP/Q regulatory system and partially on the transcriptional regulator SlyA, as indicated by abolished or diminished ClyA expression in a *phoP* or *slyA* mutant (Fig. 1b, c). The second gene on the SPI-18 operon encodes a putative invasin named TaiA[6]. We found that, like ClyA, TaiA chromosomally epitope-tagged with a 3xFLAG, is also strongly induced within HeLa epithelial cells as well as during growth in low $Mg^{2+}$ medium (supplementary Fig. 1a).

We were further interested in how quickly ClyA expression occurs in bacteria within infected host cells. Therefore, we used immunofluorescence microscopy, a more sensitive method to detect ClyA expression in bacteria. Wild-type *S*. Paratyphi A, Δ*phoP* or Δ*slyA* mutants expressing chromosomally encoded ClyA:3xFLAG, were used to infect HeLa epithelial cells and PMA-differentiated THP-1 macrophages. After the indicated time points, the infected cells were collected and treated with lysozyme. This treatment allowed for the detection of total ClyA within the bacteria. Images were captured by immunofluorescence microscopy from randomly selected fields in both the red (LPS signal to identify all bacterial cells) and green (to identify ClyA-positive bacteria) channels. The number of bacterial cells positive for green fluorescence signals (ClyA) was quantified relative to the total number of bacterial cells analyzed (LPS). A total of 10 regions of interest (ROIs), each containing approximately 20 cells, were analyzed for each cell type. The immunofluorescence microscopy analyses revealed strong ClyA expression occurring as early as 2 h post-infection in both epithelial and macrophage cells (Fig. 1d and Supplementary Fig. 1c). At this time point, up to 85% of intracellular wild-type bacteria already expressed ClyA in epithelial cells, while 70% of intracellular bacteria were ClyA-positive in THP-1 macrophages. Over time, the population of ClyA-positive bacteria increased slightly, reaching a maximum of 95% ClyA-positive intracellular bacteria after 24 h post-infection in HeLa and THP-1 cells (Fig. 1d). However, at very early time points, such as immediately after invasion or 1 h post-invasion, the percentage of ClyA-expressing bacteria remained low. Consistent with the results illustrated in the Western Blot analysis of Fig. 1b, the mutation of *phoP* or *slyA* significantly reduced the expression of ClyA within both infected cell types at all measured time points (see Supplementary Fig. 1b).

Next, we were interested in whether bacteria residing within the SCV or within the cytoplasm of infected host cells express ClyA. Therefore, we constructed two types of reporter strains. Both reporter strains express chromosomally encoded 3xFLAG epitope-tagged ClyA. One reporter strain additionally harbors a plasmid that expresses a red fluorescence protein (RFP-T) under the control of the *ssaG* promoter. The gene product of *ssaG* is part of the SPI-2 Type 3 Secretion System apparatus of *Salmonella* that is solely induced within the SCV[25]. The other reporter strain expresses a plasmid with RFP-T under the control of the *uhpT* promoter that is specifically induced in the glucose-6-phosphate-containing environment of the host cell cytoplasm and inactive within the glucose-6-phosphate-lacking SCV. Both promoter constructs, used as reporters for fluorescence microscopy studies of precise intracellular localizations of *Salmonella* within infected cells, have been successfully established in previous work[26,27]. In our study, we conducted colocalization analysis of green fluorescence signals indicating ClyA expression and red fluorescence signals representing the corresponding promoter RFP fusion constructs. The colocalization analysis using the *ssaG* promoter (P-*ssaG*) revealed that at 4 h and 24 h post-infection, approximately 95% to 98% of ClyA green-positive bacteria also expressed RFP driven by the intravacuolar cues-induced *ssaG* promoter (Fig. 1e, f). In contrast, only 2–3% of ClyA-positive bacteria exhibited RFP expression driven by the cytoplasmic induced *uhpT* promoter (Fig. 1e, f). These results clearly demonstrate that nearly all ClyA-expressing bacteria reside within the SCV. Therefore, ClyA of *S*. Paratyphi A can be considered an intravacuolar-induced cytotoxin. The induced expression of ClyA within the SCV did not result in the escape of bacteria from the SCV into the host cytoplasm, since the ratio of cytosolic bacteria compared to the total intracellular bacteria remained consistent in cell infected with the wild-type bacteria (34% cytosolic) and ClyA-negative Δ*clyA* mutants (31% cytosolic), respectively (Supplementary Fig. 1d). In summary, we have established that ClyA of *S*. Paratyphi A is an exclusively intravacuolar-induced cytolysin controlled by the PhoP/Q two-component regulatory system and the transcriptional regulator SlyA. Its expression occurs in the majority of intracellular bacteria as early as 2 h post-infection.

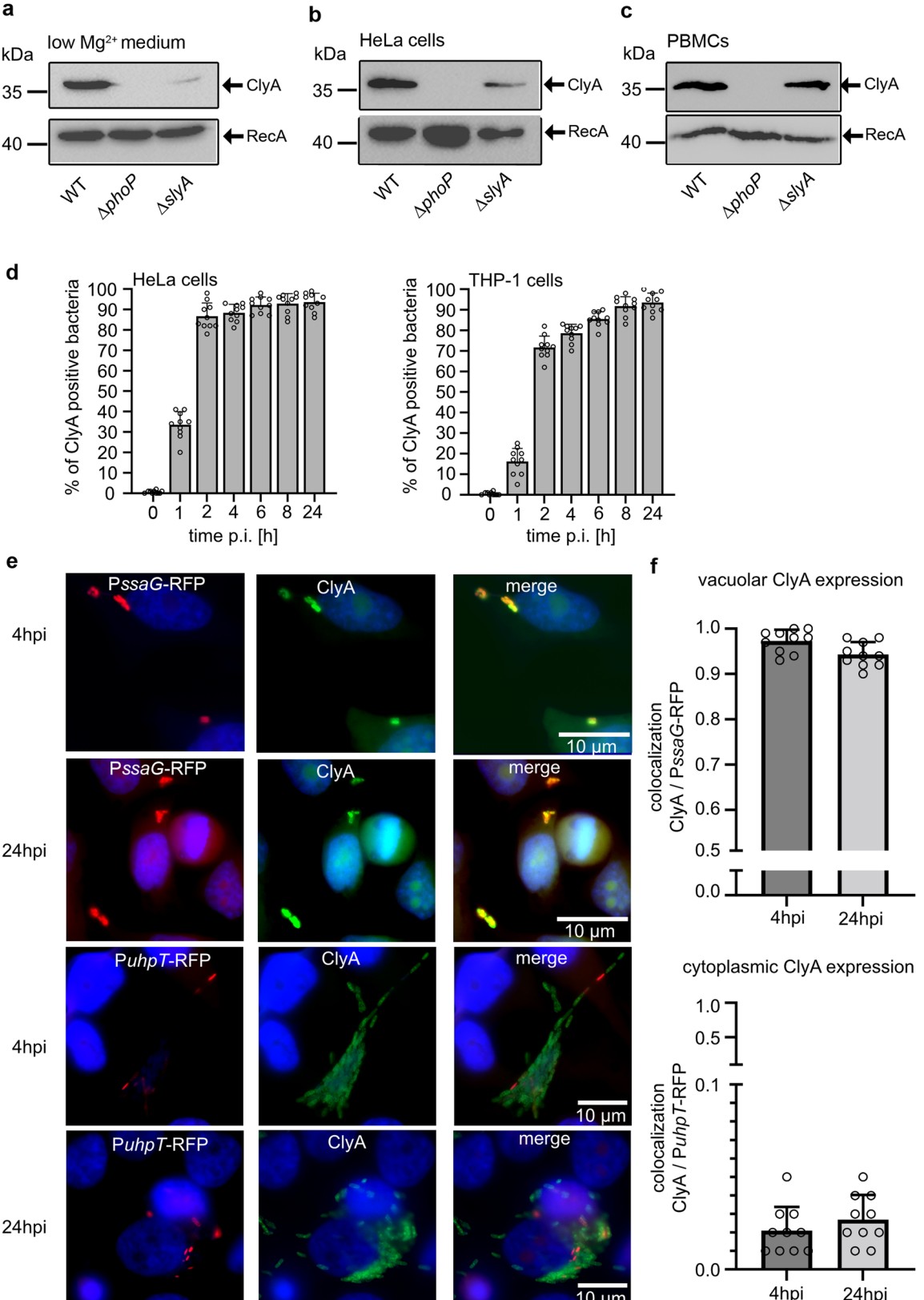

The impact of ClyA and TaiA on the invasion of host cells and the intracellular replication frequency of *S.* Paratyphi A

Previous studies on ClyA and TaiA (Typhi-associated invasine A) in *S.* Typhi have indicated their involvement in host cell invasion and bacterial survival within infected host cells[6,11]. Both genes are located on the same pathogenicity island, SPI-18, and are co-expressed

under the same regulon. Therefore, in this study, we compared the invasion rates and intracellular survival of wild-type *S.* Paratyphi A with clean deletion mutants in Δ*clyA*, Δ*taiA*, and a clean double mutant Δ*clyA*/Δ*taiA* using gentamicin protection assays. Tendentially, the *taiA* mutant exhibited lower invasion rates in HeLa epithelial cells (1.7% of inoculum), THP-1 macrophages (4% of

**Fig. 1 | ClyA expression in low Mg²⁺ medium and infected host cells.** For Western blot analyses, ClyA-3xFLAG was detected using an α-FLAG antibody and a secondary α-mouse-HRP antibody. **a** Bacterial cultures of chromosomally epitope-tagged ClyA:3xFLAG in wild-type, *ΔphoP,* and *ΔslyA* were grown in low Mg²⁺ medium for 24 h. **b** HeLa cells and (**c**) PBMCs were infected (MOI 100) with the same strains as mentioned before. After 24 hpi for the HeLa cells and 2 hpi for the PBMCs, host cells were lysed and intracellular bacteria were harvested. The arrowheads indicate ClyA:3xFLAG (36.5 kDa) in the upper panel and RecA (42 kDa) as a loading control in the lower panel. Western blot analyses were repeated with three independent biological replicates. Source data are provided as a Source Data file Blots.
**d** Quantification of HeLa and THP-1 cells infected with wild-type *S*. Paratyphi A with epitope-tagged ClyA:3xFLAG. Cells were infected with MOI 30, fixed at the indicated timepoints, and stained for *S*. Paratyphi using α-LPS (red fluorescence) and α-ClyA:3xFLAG antibodies (green fluorescence). DAPI was used for DNA staining. By immunofluorescence microscopy, a total of 10 regions of interest (ROIs), each

containing 20 host cells ($n = 200$), were quantitatively analyzed for each cell type. The percentage of green-positive bacteria (ClyA:3xFLAG) against the total number of red-stained bacteria (LPS) was determined and plotted. Each column represent the mean of 10 ROIs (dots). Data are presented as mean values ± SD. Source data are provided as a Source Data file Graphs (**e**) Immunofluorescence microscopy images of HeLa cells infected with *S*. Paratyphi A reporter strains encoding for RFP under the control of a *ssaG* (intravacuolar) or *uhp-T* (cytoplasmic) promoter and chromosomally tagged ClyA:3xFLAG. HeLa cells were infected with MOI 30, fixed after 4 or 24 h and stained for ClyA:3xFLAG. **f** Quantification of colocalization of *ssaG* or *uhp-T* promoter driven RFP activity (red fluorescence signal) and ClyA:3xFLAG (green fluorescence signal) expressing bacteria. 10 randomly selected ROIs each containing approximately 20 cells ($n = 200$) were quantified as described before. Data are presented as mean values ± SD. Source data are provided as a Source Data file Graphs.

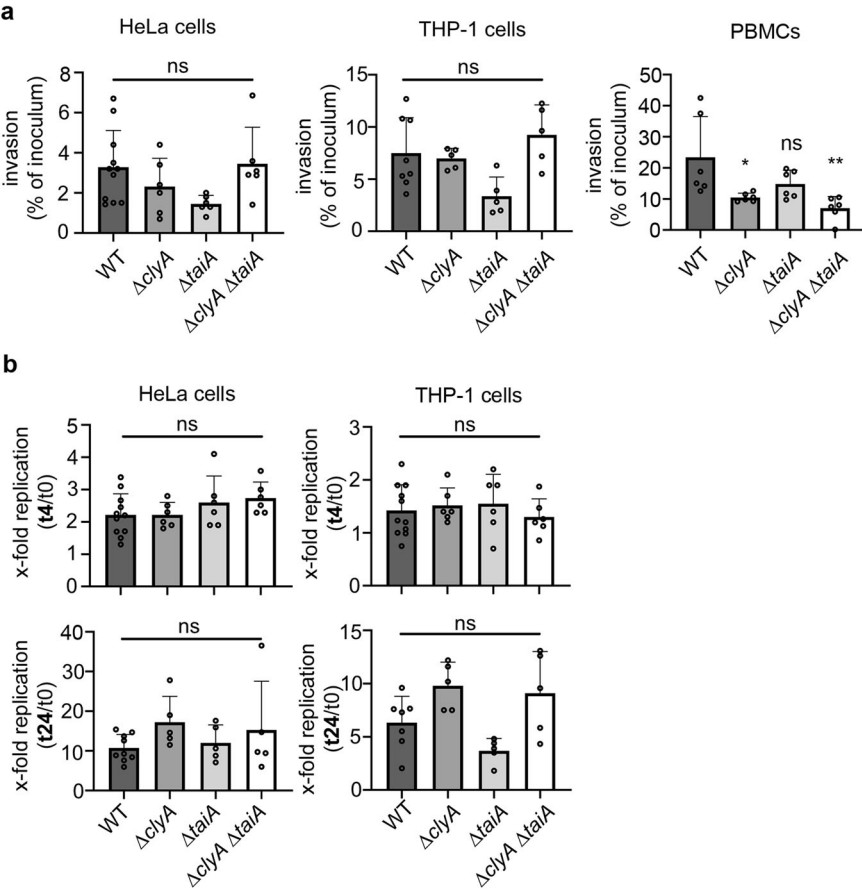

**Fig. 2 | Impact of ClyA and TaiA on host cell invasion and intracellular replication. a** To determine the host cell invasion rate, HeLa, THP-1 cells, and isolated PBMCs were incubated with *S*. Paratyphi A wild-type, *ΔclyA*, *ΔtaiA* mutants, and a *ΔclyA/ΔtaiA* double mutant (MOI 20) for 1 h. The invasion rate, expressed as a percentage of the inoculum, was determined using a gentamicin protection assay. Statistical analysis was performed using Ordinary one-way ANOVA and Šídák's multiple comparisons test. Not significant (ns): $p > 0.05$, *$p = 0.016$, **$p = 0.002$. **b** To assess the intracellular replication rate, HeLa and THP-1 cells were infected

with *S*. Paratyphi A wild-type, *ΔclyA*, *ΔtaiA* mutants, and *ΔclyA/ΔtaiA* double mutant (MOI 20). The replication rate after 4 h (t4) or 24 h (t24) incubation was determined relative to the number of bacteria present at the time of uptake (0 h). For all experiments ≥ five independent biological replicates were analyzed. Statistical analysis was performed using Ordinary one-way ANOVA and Šídák's multiple comparisons test. Not significant (ns): $p > 0.05$. Data are presented as mean values ± SD. Source data are provided as a Source Data file Graphs.

inoculum), and isolated PBMCs (14.8%). However, despite repeated experiments, the differences compared to wild-type *S*. Paratyphi A (3.5% for HeLa, 7% for THP-1, and 23.4% for PBMCs) were not statistically significant (Fig. 2a). For the *ΔclyA* mutant, significantly less invasion was detected only in isolated primary immune cells. In contrast, for epithelial cells and macrophages, only a non-significant trend was observed. The same findings applied to the *ΔclyA/ΔtaiA* double mutant (Fig. 2a).

The intracellular replication frequencies of wild-type *S*. Paratyphi A compared to *ΔclyA*, *ΔtaiA*, and *ΔclyA/taiA* mutants showed no significant differences in HeLa epithelial cells or THP-1 macrophages at 4 h and 24 h post-infection (Fig. 2b). To expand the screening for invasion and replication impacts, human intestinal epithelial cells such as Caco-2 and HT29-MTX were included in the study. However, neither cell type exhibited significantly different invasion or replication rates for the wild-type compared to the *clyA* or *taiA* mutant strains

(Supplementary Fig. 2a, b). Since isolated PBMCs were unstable for longer incubation times, they were not included in replication analyses.

Additionally, we analyzed the expression of ClyA before and during the invasion process by Western blot analysis. Detectable amounts of ClyA were not observed either before or during the 1-h contact with the host cells (invasion process). Moreover, immediately after invasion (0 hpi) or early after infection (1 hpi), ClyA could not be detected in the infected host cells (Supplementary Fig. 2c). Therefore, we conclude that only very small amounts of ClyA and TaiA, below the detection limit of Western blot analysis, are present during the invasion process. These minimal levels appear to affect invasion into isolated primary immune cells, but do not significantly impact invasion into stable epithelial cells or THP-1 macrophages. Furthermore, the intracellular expression of ClyA at later time points (4 h and 24 h) does not affect the replication rate of *S*. Paratyphi A within HeLa epithelial cells, THP-1 macrophages, or intestinal epithelial cells.

## A peptidoglycan hydrolase-dependent Type 10 secretion system facilitates the secretion of ClyA from intracellular bacteria

For ClyA to exert its activity as a pore-forming cytolysin on potential target cells, it must be released from the bacteria. Surprisingly, little is known about the mechanism of ClyA secretion by typhoidal *Salmonella*, except for its detection within the bacterial periplasm and visualization within outer membrane vesicles (OMVs) through ClyA overexpression studies[5,28]. Since ClyA is exclusively produced within infected cells, we hypothesized that its secretion depends on a system active within the SCV. In *S*. Typhi, the TtsA peptidoglycan hydrolase-dependent T10SS has been shown to be active in the SCV, mediating the secretion of intravacuolar expressed typhoid toxin[20]. To test the hypothesis that the same secretion system is responsible for ClyA secretion, we analyzed ClyA detection in *S*. Paratyphi A wild-type or Δ*ttsA* mutant infected host cells using immunofluorescence microscopy. We found that, unlike the wild-type, ClyA could not be detected in *S*. Paratyphi A Δ*ttsA* mutant-infected cells (Fig. 3a). Treatment of these cells with an exogenously added peptidoglycan hydrolase (lysozyme) completely recovered the otherwise absent ClyA fluorescence signals of the Δ*ttsA* mutant, proving that ClyA is still expressed in the mutant strain. Quantification of ClyA signals over time revealed that, over 24 h post-infection, no ClyA signals could be detected in Δ*ttsA* mutant-infected HeLa epithelial cells as well as THP-1 macrophages (Fig. 3b and Supplementary Fig. 3). In contrast, wild-type infected HeLa and THP-1 cells showed increasing signals of ClyA-secreting bacteria over time, consistent with previous studies on the TtsA-dependent secretion of typhoid toxin in *S*. Typhi where TtsA-dependent detection of typhoid toxin similarly increased over time[19,20].

To characterize the subcellular compartment from which ClyA is released into the extracellular space, we investigated its subcellular location in bacterial within infected HeLa epithelial cells using Western blot analyses. As expected, we found ClyA detectable in the bacterial cytoplasm of wild-type and Δ*ttsA* mutants (Fig. 3c). Both strains also contained significant amounts of ClyA within the periplasm, akin to the well-known periplasmic localized maltose-binding protein (MBP). Importantly, RecA, used as an exclusively cytoplasmic control protein, was not detected within the periplasmic fraction, ruling out cross-contamination by cytoplasmic proteins. Interestingly, within the host cell cytoplasm and in the host cell supernatants, ClyA could only be detected in wild-type infected cells (Fig. 3c). In contrast, a Δ*ttsA* mutant showed no detectable ClyA in neither the cytoplasm nor the supernatant of infected cells. Notably, the periplasmic control protein MBP, but not the cytoplasmic control protein RecA, could be detected in the host cell cytoplasm. This indicates that active secretion of periplasmic-localized proteins by the TtsA-dependent T10SS occurs rather than the lysis of bacterial cells and the passive release of bacterial proteins.

As mentioned previously, an in vitro study has demonstrated that *S*. Typhi, grown in bacterial culture medium and induced to over-express ClyA from a plasmid, secretes ClyA via the release of bacterial OMVs[28]. Therefore, we were interested in determining whether this phenomenon also occurs for intracellular secretion of ClyA from *S*. Paratyphi A or under conditions mimicking the intravacuolar environment using low $Mg^{2+}$ medium. THP-1 macrophages were infected with wild-type *S*. Paratyphi A expressing chromosomally encoded 3x FLAG-tagged ClyA. Following incubation under gentamicin (10 μg/ml) for 24 h, the infected host cells were washed to remove dead extracellular bacteria. Subsequently, the infected cells were lysed to retrieve the host cytosol, including bacteria. The host cell cytoplasm was then centrifuged at a standard speed of 10,000 $g$ for 10 min at 4 °C. The resulting pellet containing bacteria and host cell components was collected. The supernatant was filtered through a 0.45 μm filter, and a 100 μl sample was removed for further Western blot analysis, while the remainder was subjected to ultracentrifugation (120,000 $g$ for 2 h) to examine whether it contained soluble or membrane-bound ClyA. The ultra-supernatant and the retrieving ultra-pellet were carefully collected and used for Western blot analysis. The analysis indicates that standard centrifugation results in ClyA being present in both the pellet and the supernatant (Fig. 3d). The pellet likely contains non-secreted ClyA still within bacteria, while the supernatant already shows released ClyA from the bacteria. Upon ultracentrifugation of this cell-free supernatant, the analysis clearly demonstrates that only membrane-bound ClyA, found in the pellet, can be detected (Fig. 3d). Soluble ClyA proteins that would have ended up in the supernatant after ultracentrifugation could not be detected. When bacteria were grown for 24 h in the low $Mg^{2+}$ medium the same centrifugation steps were employed. Standard centrifugation could detect ClyA in both the pellet and the cell-free supernatant (Fig. 3e). In contrast, ultracentrifugation of the cell-free supernatant revealed that ClyA was found in the pellet, suggesting that the majority of ClyA is bound to or within a membrane (Fig. 3e).

## ClyA of *S*. Paratyphi A exhibits specific cytolytic activities towards macrophages and erythrocytes

A study on *S*. Typhi has demonstrated that ClyA of *S*. Typhi, when heterologously overexpressed in *E*. *coli*, induces the lysis of erythrocytes upon contact with the bacteria[5]. In the same study, a *S*. Typhi strain Ty21a also exhibited a small lysis zone on blood agar plates, dependent on the presence of *clyA*. Therefore, we were interested in investigating whether ClyA of *S*. Paratyphi A demonstrates similar features against erythrocytes and also included other human cell types such as epithelial cells and macrophages in the examination. We employed contact hemolytic assays using whole horse and human blood, alongside highly sensitive luminescence assays, to assess the lysis of various potential target cells. We measured the release of cytoplasmic lactate dehydrogenase (LDH) from human epithelial cells such as HeLa, Caco-2, and HT-29 MTX cells, as well as THP-1 and U937 macrophage-like cell lines upon exposure to cell-free bacterial supernatants. To artificially induce ClyA production during bacterial growth, we employed wild-type bacteria expressing plasmid-encoded ClyA or SlyA under a rhamnose-inducible promoter in LB medium supplemented with rhamnose. Negative controls included wild-type bacteria, or a clean Δ*clyA* deletion mutant with or without empty vectors. For complementation, plasmid-encoded ClyA under a rhamnose-inducible promoter was used in the Δ*clyA* deletion mutant. The hemolytic assays clearly showed that cell-free supernatants from wild-type *S*. Paratyphi A, grown under standard LB-rich conditions without rhamnose-induced expression of *clyA*, did not produce significant amounts of ClyA in the supernatants to lyse neither human nor horse erythrocytes (Fig. 4a and Supplementary Fig. 4b). In contrast, induced expression of *clyA* or *slyA*, the well-described transcriptional activator of *clyA*, resulted in severe lysis of erythrocytes within the first 2 h upon contact

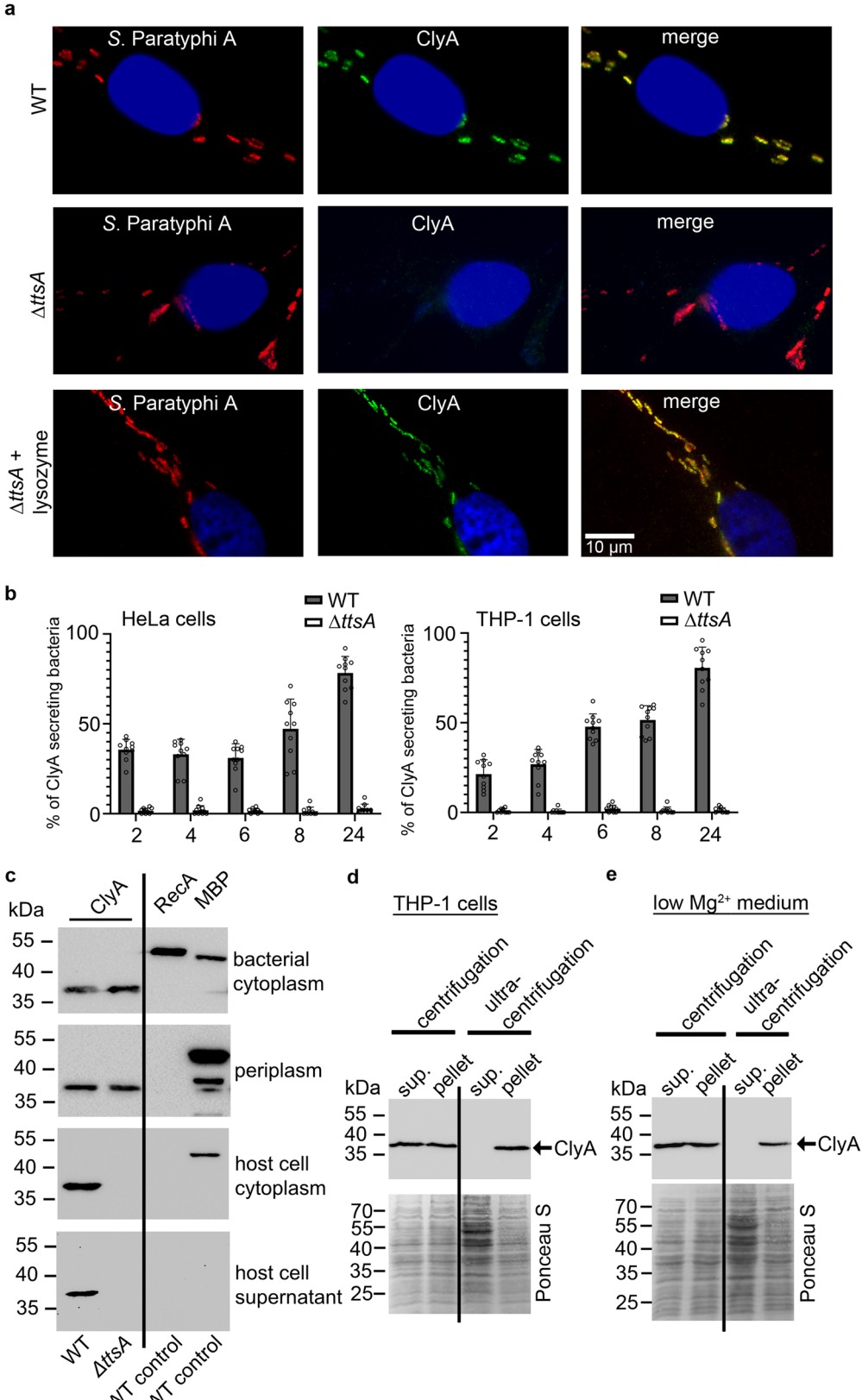

with the supernatants. Using the supernatants of rhamnose-induced *clyA*-expressing wildtype and complemented Δ*clyA* mutant bacteria on THP-1 and U937 macrophages, resulted in a significantly increased release of LDH after 4 h of incubation for THP-1 cells and after 2 h of incubation for U937 macrophages compared to Δ*clyA* mutants and wild-type bacteria with empty vector (Fig. 4b, c). The same experiment

was performed with HeLa epithelial cells, polarized and non-polarized Caco-2, and HT-29 MTX intestinal epithelial cells. Interestingly, all tested epithelial cells exhibited lysis resistance towards ClyA, as indicated by no significant differences in LDH releases upon contact with supernatants of *clyA*-expressing bacteria and Δ*clyA* mutants (Fig. 4d–f and Supplementary Fig. 4c).

**Fig. 3 | Secretion of ClyA by the TtsA peptidoglycan hydrolase-dependent Type 10 secretion system. a** Immunofluorescence microscopy images of HeLa cells infected with *S.* Paratyphi wild-type or Δ*ttsA* mutant strain with epitope tagged ClyA:3xFLAG (MOI 30). After 24 hpi the cells were harvested and stained for *S.* Paratyphi A using an α-LPS antibody and for ClyA:3xFLAG using an α-FLAG anti-body. As secondary antibodies Alexa Fluor™ 594 goat anti-rabbit and Alexa Fluor™ 488 rabbit anti mouse were used. For DNA staining, DAPI was used. If indicated, lysozyme (100 μg/ml, Sigma Aldrich) was added. **b** The percentage of ClyA-secreting and, consequently, ClyA:3xFLAG-positive bacteria (green fluorescence) was quantified relative to the total bacterial population (LPS staining, red fluorescence), within infected HeLa cells (MOI 30). A series of 10 regions of interest (ROIs) were selected for this analysis. Each ROI encompassed approximately 20 host cells (*n* = 200) and was examined quantitatively at the specified time points. Data are presented as mean values ± SD. Source data are provided as a Source Data file Graphs (**c**) Subcellular ClyA localization in intracellular *S.* Paratyphi A. HeLa cells

were infected with *S.* Paratyphi A wild-type or Δ*ttsA* mutant expressing chromo-somally epitope tagged ClyA:3xFLAG. The subcellular fractions were collected as described in the methods and analyzed for ClyA detection via Western blot analyses. Periplasmic maltose-binding protein (MBP:3xFLAG) and cytoplasmic RecA (RecA:3xFLAG) were used as cross-contamination controls. Ultracentrifugation to detect membrane-associated ClyA in (**d**) infected THP-1 cells or during growth in (**e**) low Mg²⁺ medium. Infected THP-1 cells (MOI 100), incubated for 24 h, were gently lysed, and the samples were centrifuged at a standard speed of 10,000 *g* for 10 min to pellet bacteria and cellular components. The resulting supernatant was filtered (0.22 μm) and subjected to ultracentrifugation. For growth in low Mg²⁺ medium, bacteria were pelleted by standard centrifugation, and the filtered cell-free super-natant was used for ultracentrifugation. Standard Ponceau staining of the same blots was used to visualize protein contents. **c–e** Western blot analyses were repeated with three independent biological replicates. Source data are provided as a Source Data file Blots.

The in vitro assays conducted with artificially induced ClyA pro-duction revealed that ClyA specifically targets macrophages but is inactive against epithelial cells. However, given our observation of high ClyA expression within infected epithelial cells, we hypothesized that infected epithelial cells, which are insensitive to ClyA-mediated cell lysis, may naturally release significant amounts of ClyA into the extracellular space. To further investigate this hypothesis, we infected HeLa cells with wild-type *S.* Paratyphi A, a clean Δ*clyA* deletion mutant, and the complemented Δ*clyA* deletion mutant. After 24 h post-infec-tion, cell-free HeLa cell supernatants were collected and used for LDH release assays with THP-1 macrophages, as described previously. Intriguingly, wild-type infected HeLa cells produced and accumulated significant amounts of ClyA in their supernatants, leading to rapid LDH release from THP-1 macrophages (Fig. 4g). In contrast, supernatants from Δ*clyA*-infected HeLa cells did not significantly increase LDH release from THP-1 cells over time. This effect could be reversed by using the complemented Δ*clyA* mutant strain. Our results clearly demonstrate that ClyA of *S.* Paratyphi A exhibits features of an active but cell type-specific cytolysin. We posit that *S.* Paratyphi A-infected epithelial cells represent natural ClyA producer cells that release ClyA into the extracellular space to lyse macrophages in close proxi-mity (Fig. 5).

## Discussion

In contrast to non-typhoidal *Salmonella enterica* serovars like *S.* Typhimurium, known for infecting a broad range of hosts and typically causing self-limiting gastroenteritis[29,30], typhoidal *Salmonella* such as *S.* Typhi and *S.* Paratyphi A are exclusive human pathogens, eliciting severe systemic illness[1,31–33]. Genome analyses, particularly in compar-ison with *S.* Typhimurium, have yielded valuable insights into their evolutionary history. Typhoidal *Salmonellae* appear to have relatively recently entered the human population, possessing few unique genes in contrast to non-typhoidal serovars, most of which are linked to lysogenic bacteriophages or genomic islands, and fewer encoding potential virulence factors[2,4,34,35]. In our study, we focused on ClyA, a pore-forming cytolysin encoded on SPI-18, along with TaiA, described as an invasion protein[6]. These virulence factors are found in *S.* Typhi and Paratyphi A but are largely absent from non-typhoidal *Salmonella* serovars such as *S.* Typhimurium. Our investigations revealed that ClyA and TaiA of *S.* Paratyphi A are strongly expressed within infected host cells and that ClyA expression relies on functional regulators such as PhoP/Q and SlyA (Fig. 1a–d and Supplementary Fig. 1a, b). Similar to typhoid toxin, its expression is contingent on *S.* Paratyphi A being located within the SCV (Fig. 1e–f and Supplementary Fig. 1c). Only a very small, insignificant population showed ClyA expression within the host cell cytoplasm, likely resulting from bacteria that had recently escaped the SCV. Our findings align with studies by other groups demonstrating the high dependence of *clyA* expression on the PhoP/Q two-component regulation system and transcriptional regulator

SlyA[6,9,14], both significantly induced by environmental cues within the SCV[23,36]. When compared to the expression patterns of typhoid toxin in *S.* Typhi, we observed robust ClyA expression as early as 2 h post infection, contrasting with the slower induction of typhoid toxin, which requires up to 24 h for strong expression[19,20,37]. This suggests that intracellularly produced ClyA plays a crucial role earlier in the *Salmonella* infection process compared to typhoid toxin. However, it is vital to acknowledge that while Paratyphi A and Typhi are closely related, they remain evolutionarily distinct, exhibiting differential behavior within infected host cells[38,39]. Consequently, future research efforts should encompass investigations into ClyA in *S.* Typhi. In this context, a previous study has demonstrated that *S.* Paratyphi A exits the SCV in great numbers, establishing residence within the host cell cytoplasm[24]. In light of ClyA being a membrane-active pore-forming toxin, we explored the potential influence of ClyA on membrane dis-ruption of the SCV and the subsequent release of bacteria into the host cell cytoplasm. A comparative analysis between the wild-type and a clean Δ*clyA* deletion mutant revealed no differences in the quantities of cytoplasmic bacteria (Supplementary Fig. 1d). Consequently, we posit that ClyA does not impact the integrity of the SCV membrane and, as a result, does not contribute to the increased exit of *S.* Para-typhi A into the host cell cytoplasm.

Adapted to an intracellular lifestyle, *Salmonella* expresses multi-ple proteins facilitating invasion into host cells and promoting survival or replication within them. Previous investigations on *S.* Typhi regarding the impact of ClyA on invasion and intracellular survival have yielded conflicting results. Faucher et al. reported a slight but statistically significant increase in bacterial uptake in HeLa epithelial cells upon *clyA* deletion, while Fuentes et al. observed a significant decrease in the uptake of Δ*clyA* mutants into Hep2 epithelial cells[6,11]. In our study focusing on *S.* Paratyphi A, we observed significant differ-ences in the invasion of blood-isolated primary PBMCs (Fig. 2a). However, we only noted statistically insignificant tendencies of reduced invasion by the Δ*clyA* mutant in epithelial cells and THP-1 macrophages (Fig. 2a and Supplementary Fig. 2a). Whether these variations from previous studies stem from minor activity differences between ClyA of *S.* Paratyphi A and *S.* Typhi requires further investi-gation in future studies. Nevertheless, considering that strong ClyA expression was detected only under intravacuolar conditions (Fig. 1b–f) and that ClyA amounts in bacteria outside of host cells are below the detection limit of Western blot analyses (Supplementary Fig. 1a and Supplementary Fig. 2c), we posit that these minor amounts of ClyA are only sufficient to aid in invasion into sensitive, professional phagocytes found in isolated PBMCs. To invade epithelial cells and THP-1 macrophages, these small amounts of ClyA are not sufficient to impact the invasion properties of *S.* Paratyphi A. Regarding the impact of TaiA on invasion, our results are consistent with a previous study, which showed that deletion of *taiA* had no effect on invasion, only its overexpression[6]. Once inside invaded host cells, *Salmonella*

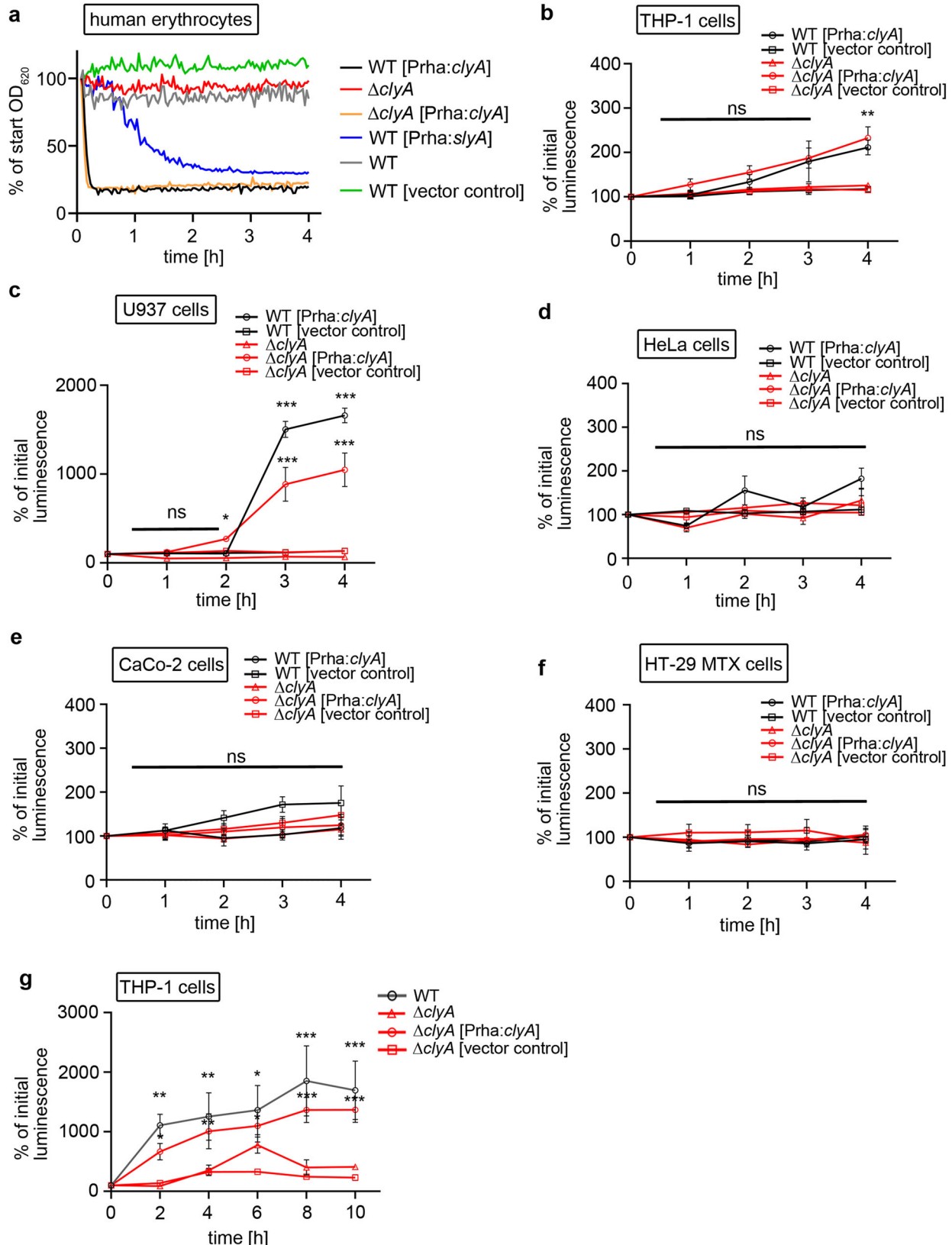

establishes residence within the SCV. We assessed whether intracellular ClyA expression has an impact on the survival and replication of *S.* Paratyphi A within infected epithelial cells or THP-1 macrophages (Fig. 2b and supplementary Fig. 2b). Consistent with published data, we observed a tendency toward slightly enhanced survival in THP-1 macrophages for the Δ*clyA* mutant compared to wild-type bacteria,

although the differences in our experiments were not statistically significant[6]. Even using human epithelial cells such as HeLa, Caco-2, and HT-29 MTX, no significant differences were detected in intracellular survival and replication.

ClyA has been extensively studied for many years, with much of our knowledge derived from investigations on its homologous protein,

**Fig. 4 | ClyA-dependent cytolysis of various target cells.** The impact of ClyA on various host cells was analyzed by hemolysis tests and LDH-release assays. **a** Supernatants of different *S*. Paratyphi A strains (wild-type, wild-type with P*rha:slyA*, wild-type with P*rha:clyA*, Δ*clyA*, Δ*clyA* with P*rha:clyA and* wild-type with empty vector control) were collected after 24 h under plasmid- inducing conditions (LB + 0.1% rhamnose) filtered and then incubated with 0.1% human erythrocytes. Optical densities (OD$_{620}$) were measured every 2 min for 4 h in a plate reader (Clario Star). **b**–**f** Supernatants of *S*. Paratyphi A [P*rha:clyA*], Δ*clyA* mutant, complemented Δ*clyA* mutant [P*rha:clyA*], and empty vector controls grown for 24 h under plasmid-inducing conditions (LB + 0.1% rhamnose) were collected and filtered (0.45 μm). Cell-free supernatants were incubated with (**b**) THP-1 cells, (**c**) U937 cells, (**d**) HeLa cells, (**e**) non-polarized Caco-2 cells, and (**f**) HT-29 MTX cells for up to 4 h. Samples were taken every hour and LDH-release was measured in a luminescence plate reader (Clario Star). All data were acquired from three independent biological replicates. Significance was determined using 2way ANOVA Šídák's multiple comparisons tests. **b** Not significant (ns): $p > 0.05$, **: $p = 0.0015$. **c** Not significant (ns): $p > 0.05$, *$p = 0.0365$, ***$p = 0.0001$. **d**–**f** Not significant (ns): $p > 0.05$. **g** HeLa cells were infected with *S*. Paratyphi wild-type, Δ*clyA* mutant strain, complemented Δ*clyA* mutant [P*rha:clyA*], and empty vector control (MOI 30). After 24 h post-infection (24 hpi), supernatants of infected HeLa cells were collected and added to THP-1 macrophages. Samples were taken every 2 h and LDH-release was measured in a luminescence plate reader (Clario Star). All data were acquired from three independent biological replicates Significance was determined using 2way ANOVA Šídák's multiple comparisons tests. *$p ≤ 0.0379$, **$p ≤ 0.0056$, ***$p ≤ 0.0006$. Data are presented as mean values ± SD. Source data are provided as a Source Data file Graphs.

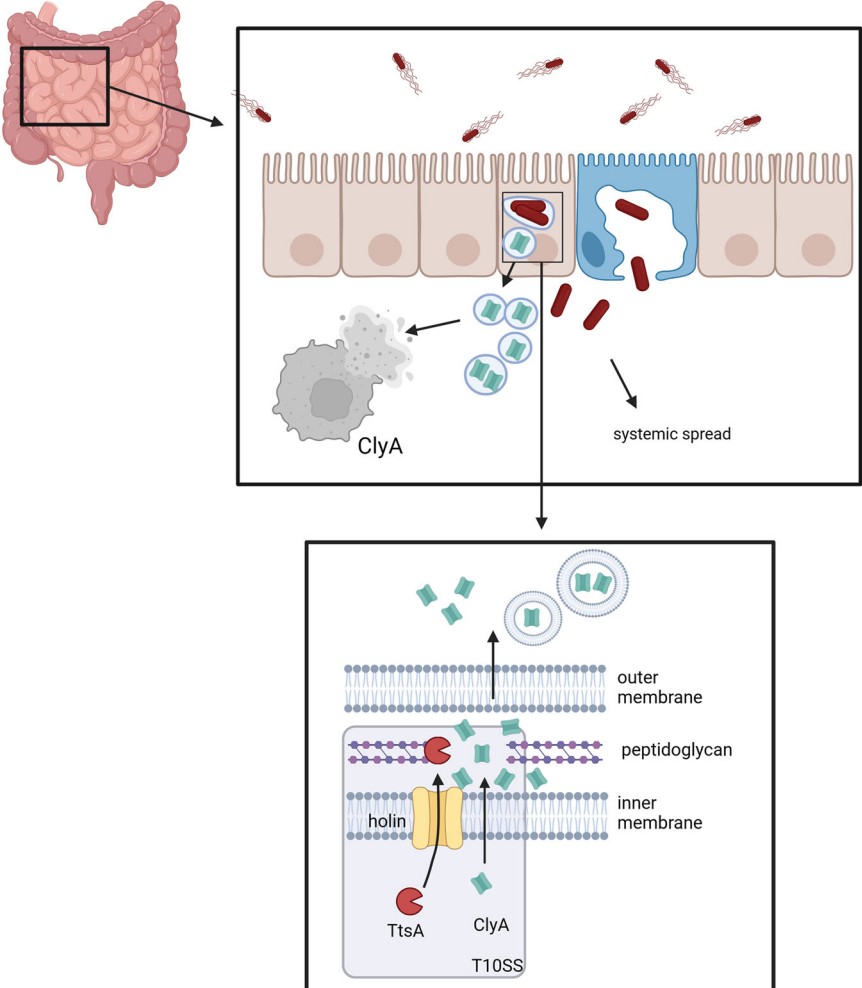

**Fig. 5 | The current model depicts the impact of ClyA on the infection process of** ***S.*** **Paratyphi A.** *S*. Paratyphi A infects epithelial cells and resides within the *Salmonella*-containing vacuole. This intravacuolar environment induces the expression of ClyA, followed by its secretion through the TtsA-dependent Type 10 Secretion System. ClyA within vesicle-membranes is then released from the infected epithelial cells, which are themselves insensitive to ClyA lysis, into the extracellular space. There, it can lyse neighboring cells sensitive to ClyA lysis. This mechanism aids the bacteria in evading the immune defenses of the human host and contributes to the characteristic systemic spread of typhoidal *Salmonella*. Created in BioRender. Geiger, T. (2024) BioRender.com/m44h507.

HlyE, from *E. coli* K12 derivatives, sharing a high amino acid sequence identity of 91%[40,41]. Studies have demonstrated that purified *E. coli* HlyE, whether in its isolated form or embedded within OMVs derived from *E. coli*, is capable of lysing erythrocytes[7,9,28]. It is crucial to note, however, that *E. coli* K12 strains are inherently extracellular bacteria, as reflected by a different expression pattern of HlyE. In *E. coli*, the expression of HlyE has been detected during growth in standard LB medium[42]. In contrast, typhoidal *Salmonella* are highly adapted to an intracellular lifestyle within host cells. This adaptation necessitates a contextual analysis of ClyA. Initial in vitro analyses with purified Typhi-ClyA and Paratyphi A-ClyA, as well as bacterial extracts from *E. coli* heterologously overexpressing typhoidal-ClyA, have demonstrated that both toxins exhibit the ability to induce pores in artificial membranes and erythrocytes, as well as lyse erythrocytes[5,9,28]. Our data, consistent with these findings, demonstrate that when homologously overexpressed for 24 h, ClyA of *S*. Paratyphi A can also be found in the

bacterial supernatant (Supplementary Fig. 4a). When these cell-free supernatants are incubated with human or horse whole blood, they effectively induce erythrocyte lysis (Fig. 4a and Supplementary Fig. 4b) similar to *E. coli*-HlyE.

Furthermore, we investigated the impact of ClyA on more complex cell types, such as epithelial cells and macrophages that *Salmonella* encounters during the process of infection. Both, cell-free bacterial supernatants and infected epithelial cell supernatants, were incubated with various epithelial cells or macrophage cell lines. Interestingly, epithelial cells, whether HeLa or intestinal epithelial cells such as Caco-2 and HT-29 MTX, proved completely resistant to ClyA-induced cytolysis (Fig. 4d–f), while two macrophage cell lines, THP-1 and U937, showed strong sensitivity to ClyA-induced cell lysis (Fig. 4b–c, g). Previous studies with bacterial cell-free supernatants have primarily been conducted in *E. coli*, demonstrating that HlyE whether purified or from bacterial extracts, can lyse HeLa cells, mouse macrophages, various human macrophages, PMNs, and monocytes[7]. Whether differences in pore-forming activities or binding properties between Paratyphi-ClyA and *E. coli*-HlyE play a role in the distinct cell-type specificity needs to be investigated in future studies. A preliminary indication comes from in vitro pore-forming studies with Typhi and Paratyphi A-ClyA on artificial membranes, indicating a Paratyphi A-specific formation behavior that is distinct from Typhi and *E. coli* HlyE characteristics[9].

Little is known about the secretion of ClyA from typhoidal *Salmonella*, especially in the context of the well-adapted intracellular lifestyle of these bacteria. Notably, ClyA lacks a canonical signal peptide sequence for Sec-System-dependent translocation. However, it has been demonstrated to accumulate in the bacterial periplasm due to an intrinsic ability to translocate across the cytoplasmic membrane[43,44]. The final release of ClyA from *S.* Typhi has been associated with a vesicle-mediated secretion mechanism[28]. OMVs, derived from bacterial outer membranes, are enriched in periplasmic proteins and are continually released from the bacterial surface[45]. So far, ClyA-containing OMVs in *S.* Typhi have only been detected and visualized in a *clyA* frameshift mutant, trans-complemented with *clyA* on a plasmid[28]. It is noteworthy to mention that this study utilized the vaccine strain Ty21a. This strain carries a mutation in the *galE* gene, leading to the accumulation of galactose derivatives in the cell and subsequent cell lysis[46]. The inherent instability of this strain likely contributes to its unique hemolytic phenotype on blood agar plates[5]. In contrast, no other *S.* Typhi strains exhibiting hemolytic phenotypes on blood agar plates have been identified so far[9]. It is conceivable that the bacterial cell instability of Ty21a might artificially lead to the formation of OMVs and the release of ClyA, as reported previously[28]. Furthermore, this strain was cultured in standard LB medium, and bacterial culture supernatants were examined for OMVs containing ClyA. However, given the natural intracellular lifestyle of typhoidal *Salmonella* and the intracellular expression of ClyA, this environment significantly differs from the conditions investigated in the study. Consequently, it remained questionable whether typhoidal *Salmonella*, localized intracellularly, genuinely secrete ClyA via OMVs. Notably, *Salmonella* resides within the SCV, characterized by a specific environment shaped by the bacteria themselves. As described earlier, ClyA is expressed in this particular environment and needs to be secreted by the bacteria. We demonstrated that *S.* Paratyphi A expresses and accumulates ClyA in the periplasm within infected cells (Fig. 3c). Interestingly, the secretion of ClyA into the host cell cytoplasm is dependent on the TtsA-dependent T10SS (Fig. 3a–c), a system previously shown to mediate typhoid toxin secretion in *S.* Typhi[19,20]. TtsA, a specialized cell-wall active peptidoglycan hydrolase, specifically cleaves modified peptidoglycan at the bacterial poles, releasing cargo proteins like typhoid toxin through the otherwise impermeable peptidoglycan layer of the bacterial cell wall[20,22]. Given its presence and activity in the periplasm, spatiotemporally correlating with ClyA, it is

consistent with our results that TtsA is also responsible for the secretion of ClyA (Fig. 3a–c), which, like typhoid toxin, accumulates in the periplasm of bacteria. Interestingly, ultracentrifugation of the cytoplasm from infected host cells revealed that ClyA is not soluble but rather exists in a vesicle- or membrane-associated form, which is found in the pellet after centrifugation (Fig. 3d). Additional experiments using a low $Mg^{2+}$ medium to mimic the SCV environment confirmed these results (Fig. 3e). The presence of membrane-associated ClyA would explain why infected THP-1 cells, which are susceptible to ClyA lysis, do not undergo significant lysis despite producing ClyA intracellularly (Fig. 2b). In this scenario, ClyA, rendered inactive while bound to or surrounded by a membrane, would not be in direct contact with the host cell. A similar vesicle packaging of exportable toxins has been described for the intravacuolar typhoid toxin of S. Typhi, which also results in the toxin producer cells not being affected by the toxin[37,47]. Whether the membranes associated with ClyA are derived from OMVs or are a result of the host SCV, as seen with typhoid toxin, needs to be visualized and investigated further in future studies. Additionally, the hypothetical contribution of TtsA to potential OMV formation and subsequent release of ClyA intracellularly by destabilizing the peptidoglycan layer through its peptidoglycan hydrolysis activity needs further investigation. Several studies have already observed that cell wall stress and conditions inducing cell wall destabilizations and modifications can induce OMV formation[45,48,49]. Phylogenetic and genetic analyses indicate that TtsA appears to be adapted from endolysins, phage-related enzymes used by bacteriophages to lyse infected bacterial cells and exit from them[50–52]. Interestingly, despite targeting peptidoglycan, TtsA activity has evolved, leaving it with specific activity towards modified peptidoglycan[20,22]. This feature, combined with its restricted localization at the bacterial poles, prevents TtsA hydrolase activity from being lytic for *S.* Typhi during the secretion process. Previous studies on the release of ClyA support the hypothesis that phage-related components such as endolysins and holins have the potential to release sufficient amounts of ClyA from lysed bacteria[53,54]. Here we show that an active, non-lytic secretion mechanism, the T10SS, adapted from phages, releases ClyA in a more sophisticated way into the host cell. As mentioned earlier, ClyA, is present in typhoidal *Salmonella enterica* serovars and shares a close homolog in *E.coli* K-12 strains. In contrast, it is largely absent in well-described non-typhoidal *Salmonella* serovars such as *Salmonella enterica* serovar Typhimurium as well as in other members of the Enterobacteriaceae such as *Citrobacter*, *Klebsiella*, *Serratia*, and *Yersinia* species[55]. Interestingly, intracellularly adapted *Shigella* species harbor only nonfunctional *clyA* copies due to frameshift mutations[55]. This leaves typhoidal ClyA as an exceptional intracellular induced cytolysin, adapted in its secretion and specific mode of action towards certain target cells to the intracellular lifestyle of typhoidal *Salmonella*.

## Methods
### Ethics statement

Human PBMCs and erythrocytes were isolated from whole, serum-depleted, anticoagulated human donor blood. PBMCs were obtained from buffy coats of three non-pooled, single male anonymous donors, while erythrocytes were isolated from the whole blood sample of a single male anonymous donor, all sourced from a commercial blood bank (biomol, Research Donors). The company provides documentation confirming that the anonymous donors gave their consent for the use of their samples in biomedical research. No research involving humans, such as clinical studies or biomedical research using personal data, was conducted. All samples used (anonymous donors, commercially available buffy coats, or whole blood samples) did not involve the collection of personal medical data. Therefore, ethical review and approval were not required, in accordance with local legislation and institutional requirements. Horse erythrocytes were

isolated from commercially available defibrinated whole horse blood (Thermo Scientific).

## Cloning/mutagenesis

The bacterial strains and plasmids used in this study are listed in S1 Table. All *Salmonella* Paratyphi A strains are derived from *Salmonella enterica* serovar Paratyphi A 45157[56]. All in-frame deletions were generated by an optimized site-directed scarless mutagenesis protocol[57]. A recombinant PCR product containing a kanamycin resistance cassette and a recognition site for the meganuclease I-SceI is integrated by a heat-induced λ Red recombinase. In a second step, the cassette is replaced with a clean-deletion PCR product. Mutants which still contain the cassette are killed by double strand breaks by the AHT-inducible meganuclease I-SceI. Insertion of 3xFLAG epitope tag into the chromosome was done with two different methods. The strain TG0031 (clyA:3xFLAG) was constructed as described by Uzzau, Figueroa-Bossi[58] who modified the classical site-directed scarless mutagenesis by Datsenko and Wanner[59]. Therefore, we first constructed pTG0051 as template plasmid for the 3xFLAG tag fused resistance cassette and integrated it into the chromosome. The cassette was removed with the assistance of plasmid pCP20. To construct the strain TG0041 (1307:3xFLAG) we used the optimized site-directed scarless mutagenesis protocol[57] and modified the template plasmid pWRG717 as described by Uzzau, Figueroa-Bossi (58). We constructed pTG0076 and used it as new template plasmid for insertion of the chromosomal 3xFLAG tag. Primers used for cloning procedures are listed in S2 Table. All plasmids used in this study were constructed using the Gibson assembly cloning strategy. All generated plasmids and strains used in this study have been verified by nucleotide sequencing.

## Bacterial culture

*S.* Paratyphi A strains were routinely cultured on standard LB agar plates or in liquid LB medium (10 g/l NaCl, 10 g/l tryptone, 5 g/l yeast extract) on a shaking platform at 37 °C. To mimic intracellular conditions specific low $Mg^{2+}$ medium[20]) was used. It is a chemically defined medium and contains K2SO4 (0.5 mM), KH2PO4 (1 mM), (NH4)2SO4 (7.5 mM), Tris Base (50 mM), Bis Tris (50 mM), casamino acids (0.1 %), KCL (5 mM), cysteine (50 µg/ml), tryptophan (50 µg/ml), glycerol (32.5 mM) and magnesium (15 µM). When appropriate, antibiotics were added to bacterial cultures (50 µg/ml kanamycin (Roth), 100 µg/ml ampicillin, or 10 µg/ml chloramphenicol (Roth)).

## Eukaryotic cell culture

The HeLa cells were grown in Dulbecco's modified eagle's medium (DMEM, high glucose, with glutamine, Gibco), supplemented with 10% fetal calf serum (FCS, Gibco). THP-1 and U937 cells were cultivated in RPMI medium (Gibco) containing 1 mM sodium pyruvate (Gibco) and 10% FCS calf serum (FCS, Gibco). Caco-2 and HT-29 MTX cells were maintained in Dulbecco's modified eagle's medium (DMEM, high glucose, with glutamine, Gibco) containing 10% FCS and 1x non-essential amino acids (100x, MEM NEAA, Gibco). For Caco-2 cells, 1 mM sodium pyruvate (Gibco) was additionally added. All cells were incubated at 37 °C in a humidified incubator with 5% $CO_2$. THP-1 cells were differentiated by incubation in RPMI medium with 50 ng/ml phorbol 12-myristate 13-acetate (PMA) for 24 h and further 24 h in RPMI medium without PMA.

## Analyses of ClyA and TaiA expression

To analyze ClyA and TaiA expression under in vitro conditions bacteria were inoculated in LB overnight. Subcultures were inoculated 1:30 in LB or low $Mg^{2+}$ medium. After 24 h incubation, bacteria were harvested and concentrated 1:50 in Laemmli buffer. The different plasmids were induced adding 0.1% rhamnose (Roth, Germany) for 3 h before harvesting.

For analysis of intracellular expression, HeLa cells were seeded in a 10 cm dish ($8.8 \times 10^6$ cells/dish at the day of infection). The bacterial strains were inoculated overnight and sub-cultured in LB containing 0.3 M NaCl to induce T3SS-SPI1 activity[60]. Subcultures were grown until reaching an OD600 of 0.9. HeLa cells were infected with $8.8 \times 10^8$ bacteria per dish (MOI 100). The high MOI was necessary to ensure an adequate number of bacteria for subsequent Western blot protein analyses which are less sensitive. The same MOI has been used in various *Salmonella* infection studies analyzing genomic content or cellular responses without notably impacting the infected host cells[61–63]. After a 1-h incubation of bacteria with host cells (bacterial uptake), the cells were washed three times with PBS. To eliminate any remaining extracellular bacteria, 100 µg/ml gentamicin was added to DMEM, and cells were further incubated for 1 h. Subsequently, cells were incubated with DMEM containing 10 µg/ml gentamicin for up to 23 h (2–24 hpi). After incubation, cells were lysed with 5 ml PBS containing 0.1% Triton-X100. The lysate was centrifuged for 5 min at $10,000 \times g$, and the pellet was resuspended in 100 µl Laemmli buffer.

## Host cell infection and immunostaining of intracellular bacteria

**Gentamycin protection assay.** Gentamicin protection assays were performed as described previously[64]. Host cells were seeded in 24-well plates (HeLa cells $2.0 \times 10^5$ cells/well, THP-1 cells $4.0 \times 10^6$ cells/well at day of infection). For the infection, bacteria were grown overnight in LB medium. Bacteria were sub-cultured (1:30) from overnight cultures in LB 0.3 M NaCl to induce SPI1-T3SS mediated invasion[60]. Subcultures were incubated until reaching an $OD_{600}$ of 0.9 on a shaking platform at 37 °C. When the bacteria reached an $OD_{600}$ of 0.9, the bacteria were diluted in DMEM. The bacteria were added to the host cells for 60 min (MOI 20). This MOI provides a sufficient number of intracellular bacteria to proceed with the following analyses. The cells were washed three times with DPBS (Gibco) to remove the bacteria. The infected cells were further incubated for 60 min with DMEM supplemented with gentamicin (100 µg/ml) to kill remaining extracellular bacteria. For determination of the number of invasive bacteria, cells were lysed afterwards. To examine the intracellular survival, medium was replaced with DMEM containing 10° µg/ml gentamicin and cells were incubated for other 3 or 23 h. After the appropriate time, host cells were lysed with 500 µl of 0.1% Triton-X100. After 10 min incubation, the cells were rigorously pipetted up and down and the lysates were collected. The wells were rinsed with 500 µl PBS, which were added to the lysates. Lysates and the initial bacteria inocula were diluted and the CFU were determined.

**Immunostaining and fluorescence microscopy.** The percentage of ClyA:3xFLAG expressing bacteria within THP-1 and HeLa cells was determined by fluorescence microscopy and immunostaining. Cells were seeded on coverslips in a 24-well plate (HeLa cells $2.0 \times 10^5$ cells/well, THP-1 cells $4.0 \times 10^6$ cells/well at day of infection). For the infection, bacteria were cultured overnight in LB medium and sub-cultured as described. For invasion, the bacteria were added to the host cells for 60 min (MOI 30). This MOI provides a sufficient amount of intracellular bacteria to proceed with the analyses of the detection of ClyA positive bacteria. Afterward, the cells were washed three times with DPBS (Gibco) to remove any remaining extracellular bacteria. This marks the start of the CFU counting process, defined as 0 h post-infection (0 hpi). The infected cells were then incubated for 1 h with DMEM or RPMI supplemented with gentamicin (100 µg/ml) to eliminate any remaining extracellular bacteria (defined as 1 hpi). The cells were either utilized immediately or the medium was replaced with medium containing 10 µg/ml gentamicin and incubated for the appropriate time (2–24 hpi). At different time points, cells were washed once with PBS and fixed with 3% paraformaldehyde (PFA) in PBS for 15 min at room temperature. Subsequently, cells were washed once with PBS and stored in PBS at 4 °C. The fixed cells were then treated with blocking

buffer (0.1% Triton, 0.3% BSA in PBS) for 1 h. If indicated, cells were treated with lysozyme (100 μg/ml) for 30 min at 37 °C in 10 mM Tris buffer, pH 8.

Thereafter, rabbit α-LPS antibody (1:400) and mouse α-FLAG antibody (1:5000) were diluted in PBS (0.1% Triton, 0.1%BSA) and added to the coverslips. After incubation at 4 °C overnight, coverslips were washed thrice and incubated for 1 h with secondary antibodies Alexa Fluor™ 594 goat anti-rabbit and Alexa Fluor™ 488 rabbit anti mouse (Invitrogen) and 0.1 μg/ml DAPI (Sigma-Aldrich) in the dark. Thereafter, the coverslips were washed three times with PBS and mounted with ProLong Glass Hard-Set Antifade Mountant (Invitrogen) on glass slides. The ratio of ClyA:3xFLAG positive (green) and LPS positive (red) bacteria at the different timepoints was determined manually and plotted as indicated. Here, 10 randomly selected ROI each containing approximately 20 cells were analyzed. The combined analysis of these 20 cells provided the mean value of 1 dot depicted in the corresponding graphs. Each column represent the mean of 10 dots or ROIs, respectively.

### Subcellular fractionation
Subcellular fractionation of proteins in the bacterial cytoplasm, periplasm, and supernatant was conducted following a previously described protocol[65]. Confluent HeLa cells were infected with either wild-type *S.* Paratyphi A or a Δ*ttsA* mutant expressing chromosomally tagged clyA:3xFLAG at a multiplicity of infection (MOI) of 1:100. This MOI provides a sufficient amount of intracellular bacteria to proceed with the following Western blot analyses. As control proteins, expressed on rhamnose-inducible plasmids, cytoplasmic RecA and periplasmic MalE (MBP) were used. The infection was carried out as outlined in the gentamycin protection assay section. Infected cells were maintained for 24 h. Subsequently, host cell supernatants were collected, filtered through a 0.45 μm filter, and the cell-free host cell supernatants were protein-precipitated with Trichloroacetic acid (TCA) for later use in Western blot analyses. The remaining cells were lysed with 0.1% Triton X-100 treatment for 20 min. The cells, including bacteria, were then centrifuged, and the resulting supernatant, reflecting the cytoplasm of the cells, was collected, filtered, and precipitated via TCA, as mentioned earlier. The pellet, containing host cell debris and intracellular bacteria, underwent treatment with a subcellular fractionation protocol published by G. Malherbe et al.[65].

### Ultracentrifugation for the detection of membrane-associated ClyA
To determine the presence of ClyA in infected THP-1 cells, PMA-differentiated THP-1 cells (as previously described) were infected with wild-type *S.* Paratyphi A expressing chromosomally encoded 3x FLAG-tagged ClyA at a MOI of 1:100. This MOI ensures an adequate amount of intracellular bacteria for subsequent Western blot analyses. After incubation with gentamicin (10 μg/ml) for 24 h, the infected host cells were washed to remove dead extracellular bacteria. Subsequently, the infected cells were gently lysed using 0.01% Triton X-100 for 10 min to extract the host cytosol, along with bacteria. The resulting mixture was centrifuged at 10,000 g for 10 min at 4 °C, yielding a pellet containing bacteria and host cell components. The supernatant was filtered through a 0.45 μm filter, and a 100 μl sample was retained for further Western blot analyses, while the rest underwent ultracentrifugation (120,000 g for 2 h). The resulting ultra-supernatant and ultra-pellet were carefully collected and used for subsequent Western blot analyses. A standard Ponceau protein staining was performed as a loading control to demonstrate protein content equivalence. The blots were subsequently de-stained before anti-FLAG antibody was applied to detect ClyA-3xFLAG. The Western blot analysis was conducted as previously described[64].

For the detection of ClyA during growth in low Mg²⁺ medium, wild-type *S.* Paratyphi A expressing chromosomally encoded 3xFLAG-tagged ClyA were cultured for 24 h in low Mg²⁺ medium, and the same centrifugation and ultracentrifugation steps were followed as described above. A standard Ponceau protein staining was performed as a loading control to demonstrate protein content equivalence. The blots were subsequently de-stained before anti-FLAG antibody was applied to detect ClyA-3xFLAG. The Western blot analysis was conducted as previously described[64].

### Chloroquine assay
To assess the proportion of vacuolar and cytosolic bacteria, we conducted a modified chloroquine killing assay, following a previously published protocol[66]. HeLa cells were seeded in 24-well plates ($2 \times 10^5$ cells/well at the day of infection). For the infection, bacteria were grown overnight in LB medium. Bacteria were sub-cultured (1:30) from overnight cultures in LB 0.3 M NaCl to induce SPI1-T3SS mediated invasion[60]. Subcultures were incubated until reaching an OD600 of 0.9 on a shaking platform at 37 °C. When the bacteria reached an OD600 of 0.9, the bacteria were diluted in DMEM. For each bacterial strain tested, nine wells each containing $2 \times 10^5$ HeLa cells were used. For bacterial uptake the bacteria were added to the host cells for 60 min (MOI 20). This MOI provides a sufficient amount of intracellular bacteria to proceed with the following analyses. The cells were washed three times with DPBS (Gibco) to remove remaining extracellular bacteria. The infected cells were further incubated for 60 min with DMEM supplemented with gentamicin (100 μg/ml) to kill remaining extracellular bacteria. After 1 h, 3 wells per strain were lysed with 500 μl of 0.1% Triton-X100. After a 10-min incubation, the cells were rigorously pipetted up and down, and the lysates were collected as the total amount of intracellular bacteria to determine the invasion rates of the bacteria. The media of the remaining 6 wells were replaced with DMEM containing 10 μg/ml gentamicin, and cells were further incubated for 2 h. After 2 h half of the remaining cells (3 wells per strain) were incubated with 400 μM chloroquine and 10 μg/ml gentamicin, whereas the other half was kept in 10 μg/ml gentamicin. After 1 h, all cells were lysed with 500 μl of 0.1% Triton-X100. After a 10-min incubation, the cells were rigorously pipetted up and down, and the lysates were collected. The wells were rinsed with 500 μl PBS, which were added to the lysates. All of the lysates and the initial bacteria after invasion were diluted and CFUs were determined.

### Isolation of Peripheral Blood Mononuclear Cells (PBMCs) from blood samples
PBMCs were isolated from whole, serum-depleted anticoagulated human donor blood (buffy coats; three non-pooled single anonymous donors) from a commercial blood bank (biomol, Research Donors). To ensure the sterility of the cells after isolation, cells for each donor were plated on blood agar plates under aerobic and microaerophilic conditions and examined for bacterial growth, with negative results. In addition, all anonymous donors' sera were tested for HIV- and HCV-antibodies and HBsAg, and a Treponema pallidum Hemagglutination Assay (TPHA) for *Treponema pallidum* was performed (all negative results).

Ficoll-Paque-based isolation of PBMCs was performed according to the manufacturer's protocol (Sigma Aldrich). Human peripheral blood samples (10 ml) were mixed with an equal volume of balanced salt solution (0.1% Glucose, 0.05 mM CaCl₂, 0.98 mM MgCl₂, 5.4 mM KCL, 145 mM TRIS, pH 7.6). The blood was carefully layered onto the Ficoll-Paque solution and centrifuged at 400 g for 40 min at 20 °C. The upper layer containing plasma and platelets were drawn off. Beneath the mononuclear cell layer was transferred to a sterile centrifuge tube. The cells were resuspended in the balanced salt solution and washed twice by centrifugation. The washed cells were resuspended in HBSS (Gibco, Life technologies). The isolated PBMCs were counted and immediately used for infection at a MOI of 20 for invasion and 100 for Western blot analysis.

## Western blot analyses

Western blot analyses were performed as described previously[64]. For the detection of ClyA-3xFLAG, TaiA-3xFLAG or MalE-3xFLAG (MBP) a primary mouse-α-FLAG antibody (Sigma Aldrich) and a secondary α-mouse-HRP antibody (Thermo Fisher) were used. For the detection of RecA, a primary rabbit-α-RecA antibody (Thermo Fisher) and a secondary α-rabbit-HRP antibody (Thermo Fisher) were used. Standard protocols provided by the manufacturer (Abcam) were followed for Ponceau protein staining and mild de-staining.

## Cell toxicity assays

To test the effect of ClyA on host cells, ClyA was overexpressed in the WT using the plasmid pTG86 (*Prha:clyA*). As a negative control the strain TG28 (Δ*clyA*) was used. The bacteria were grown overnight in LB. Subcultures were inoculated 1:30 in LB containing 0.1% rhamnose. The bacterial supernatant was harvested and filtered with a 0.45 μm filter after 24 h.

**Erythrocytes contact assay.** Human erythrocytes were isolated from commercially available whole, serum-depleted anticoagulated human donor blood (a non-pooled single anonymous donor) (biomol, Research Donors). To ensure the sterility of the cells after isolation, cells for each donor were plated on blood agar plates under aerobic and microaerophilic conditions and examined for bacterial growth, with negative results. In addition, all anonymous donors' sera were tested for HIV-, HCV-antibodies, and HBsAg. A TPHA for Treponema pallidum was also performed (all with negative results). Horse erythrocytes were isolated from commercially available defibrinated whole horse blood (Thermo Scientific).

To test the effect on erythrocytes, bacterial supernatant was incubated with 0.1% erythrocytes. The change of $OD_{620}$ was measured in a 96-well plate with the Clariostar plate reader (BMG Labtech). The $OD_{620}$ was measured every 2 min for 4 h at 37 °C with 600 rpm shaking during incubation.

**LDH release.** To test the effect of ClyA on HeLa, Caco-2, HT29-MTX, U937, and THP-1 cells, LDH release during incubation of cells with bacterial supernatant was measured using the LDH-Glo™ Cytotoxicity Assay (Promega). Therefore, cells were seeded in a 24 well plate (HeLa, Caco-2, HT-29 MTX cells with $2.0 \times 10^5$ cells/well, and THP-1 and U937 cells with $4.0 \times 10^5$ cells/well) at day of infection. THP-1 and U937 cells were differentiated by incubation in medium with 50 ng/ml phorbol 12-myristate 13-acetate (PMA) for 24 h and further 24 h in medium without PMA. To polarize Caco-2 cells, cells were seeded in 24 well plates ($2 \times 10^5$ cells/well) and incubated for 21 days at 37 °C with 5% $CO_2$ in a humidified incubator, changing the media every second day as described previously[64]. Cells were incubated with 250 μl bacterial supernatant diluted with 250 μl RPMI medium (THP-1 and U937 cells) or 250 μl DMEM (HeLa, Caco-2, and HT-29 MTX cells) for 4 h. Two wells were treated with the same supernatant and samples were taken immediately after addition of the supernatant and thereafter every hour from the same well. Since U937 cells were not attached well, the cells were centrifuged for 5 min at 400 x *g* before taking the samples. The samples (5 μl) were taken carefully without touching cells and immediately frozen in LDH storage buffer, diluted 1:10 (200 mM Tris-Cl (pH 7.3), 10% glycerol, 1% BSA) according to the manufacturer's protocol. As a positive control, cells were lysed with 0.5% Triton after the experiment. The samples were thawed, diluted 1:5 in storage buffer and the LDH release was measured according to the manufacturer's instructions. The luminescence (emission 545 nm) was determined with the Clariostar plate reader after 30 min incubation at RT.

To test the effect of ClyA, natively expressed in HeLa cells, HeLa cells were seeded in 10 cm dishes ($8.8 \times 10^6$ cells/dish at the day of infection) and THP-1 cells were seeded and differentiated in a 24-well plate as described above. HeLa were infected as described for

the analysis of intracellular expression with MOI 30. After killing extracellular bacteria with DMEM 100 μg/ml gentamicin, medium was replaced with 6 ml DMEM 10 μg/ml gentamicin per dish, and cells were incubated for 19 h. The host cell supernatant was harvested and added to THP-1 cells (1 ml per well). Immediately and every 2 h samples (5 μl) were taken, stored and processed as described above.

## Statistical analyses

Data were plotted using GraphPad Prism 10 software. All data were acquired from ≥ three independent biological replicates. Significance is given as follows: Not significant (ns): $p > 0.05$, *$p < 0.05$, **$p < 0.01$. ***$p < 0.001$.

## Reporting summary

Further information on research design is available in the Nature Portfolio Reporting Summary linked to this article.

## Data availability

All data generated or analyzed during this study are included in this published article and its Supplementary Information. Source data are provided with this paper.

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

## Acknowledgements
We would like to express our gratitude to Guntram Grassl for generously donating the wild-type *S*. Paratyphi A strain 45157, and to Christine Josenhans for providing the THP-1 and U937 cell lines and engaging in thoughtful discussions throughout the study. We thank Roman Gerlach for providing plasmids and strains used in this study for gene mutagenesis and gene expression strategies. This work was funded by den Stiftungen zu Gunsten der Medizinischen Fakultät der Ludwig-Maximilians-Universität München, Cluster 2, and the Deutsche Forschungsgemeinschaft (DFG) project number 523077943 to T.G., L.K., and S.M. were additionally supported by the graduate program "Infection Research on Human Pathogens@MvPI".

## Author contributions
L.K. was involved in the design, conduction, and interpretation of experiments shown in this study. S.M. was involved in the conduction of Western blot analyses, L.D.H. release assays, and C.F.U. invasion assays. T.G. was involved in the design, conduction, interpretation, and supervision of this study. L.K. and T.G. wrote the paper.

## Funding

## Competing interests
The authors declare no competing interests.
