## [Peer Review File · Nature Communications]

Cytolysin A is an intracellularly induced and secreted cytotoxin of typhoidal *Salmonella*REVIEWER COMMENTS

Reviewer #1 (Remarks to the Author):

The authors detail the expression, regulation, secretion, and activity of the cytolysin ClyA in *Salmonella* serovar Paratyphi. The manuscript is well-written and elucidates the mechanism of secretion along with cell type-specific cytolytic activity. It corroborates previously known data on ClyA in serovar Typhi.

1. To validate the model proposed in Figure 5, assessing the cell type-specific activity with additional epithelial cells and macrophages would be essential.
2. In the Western Blot presented in Figure 1, including a loading control, such as a housekeeping gene, would be valuable to confirm the observed lower expression in the mutant backgrounds. This is particularly important to ascertain whether the reduced expression is a direct result of the mutation or if these mutants exhibit decreased growth in a low Mg²⁺ medium. Furthermore, it remains unclear whether the low Mg²⁺ medium used to induce PhoP activation is the same as the intracellular medium TTIM described in the Materials and Methods section.
3. The authors assert that ClyA is 'exclusively expressed within infected host cells' (lines 261-262) and 'absent in bacteria outside of host cells' (line 294). However, it appears contradictory that ClyA overexpression in LB supernatant was utilized to assess cytolytic activity. This approach seems counterintuitive, considering the stated condition-specific expression of ClyA.
4. Why use horse erythrocytes for a human pathogen?
5. The authors should provide clarification on the distribution of ClyA in *Salmonella*. While they assert that ClyA is "exclusively found in *S. Typhi* and *S. Paratyphi A* and largely absent from non-typhoidal *Salmonella* serovars" (lines 259-260) or "entirely absent in non-typhoidal *Salmonella* serovars" (line 381), a quick Blast analysis reveals its presence in several strains of serovar Indiana, Adjame, and Rubislaw... Addressing this discrepancy is crucial for defining the distribution of ClyA in *Salmonella*.
6. Why SPI-1 inducing condition was used to infected THP-1 macrophages.

Reviewer #2 (Remarks to the Author):

The authors investigated cytolysin A (ClyA) secreted by *Salmonella enterica* serovar Paratyphi A. Similar investigations on ClyA homologs in the same context have previously been published, rendering half of the data presented in this publication not so novel. Two novel findings made are: (i) ClyA is secreted in a Type 10 secretion-dependent manner (Fig. 3), and (ii) ClyA exhibits toxicity towards THP1 cells but not the SCV membrane or epithelial cells (Fig. 4). These findings are interesting. However, the authors have not described the mechanism by which ClyA exhibits selective toxicity. Furthermore, whether these in vitro observations are relevant in vivo is lacking in this manuscript. Figure 2's findings depart from the current literature. However, the authors did not provide a mechanical explanation for the disparity; if provided, this information will be helpful in the field. Without a molecular understanding, the story is simply incomplete.

We recommend that the authors conduct additional research into the mechanism by which cytolysin A specifically targets THP-1 cells but not epithelial cells or the SCV membrane. It would also be crucial to demonstrate the same cell selectivity in vivo, as well as the functional consequences. Without these mechanisms, the data given in this study are preliminary.

Figures 1 and 3 should include representative microscope images. Currently, the displayed data consists of immunofluorescence quantification. The quantification data provided in different figures does not match. For instance, Figure 1C and Figure 3B provide contradictory data; the ClyA-

positive bacteria proportion should be comparable.

The description of "2 hours post-infection" lacks clarity regarding the starting point of the counting process after the addition of bacteria. Was the counting performed immediately after adding the bacteria? Furthermore, it is crucial to present additional data that demonstrates the generation of ClyA within 2 hours after bacterial invasion.

Fig. 2: The key finding of Fig. 2 is to demonstrate the discrepancy in the current literature. Is there an explanation? Are there any sequence differences?

Minor:

Figure 1D, bottom panel: Fig. 1C indicates that around 90% of intracellular bacteria should be ClyA-positive. However, among the more than 20 intracellular bacteria shown, all are ClyA-negative.

Figure 2B: There appear to be discrepancies in the graph in the lower panel. The graph in the lower panel should display individual values.

The approaches in Figure 3 lack detail.

Fig. 4A. Human erythrocytes appear to be more relevant than horse erythrocytes. The experiment does not include SCV-like media. The authors do not include bacteria that contain an empty vector.

Reviewer #3 (Remarks to the Author):

In the manuscript entitled "Cytolysin A: An intracellularly induced and secreted cytotoxin of typhoidal *Salmonella enterica* serovar Paratyphi", the authors have investigated the conditions for expression, localization and secretion of the cytolysin A (ClyA) produced by *Salmonella* Paratyphi infected cells. Specifically:

Expression of ClyA is induced in the culture medium with low Mg²⁺ concentration, mimicking the SVC conditions, and in host cells in a PhoP/Q and SlyA-dependent manner (Figure 1). ClyA does not have a significant impact in modulating the invasion or replication of *S. Paratyphi* in HeLa cells and differentiated macrophage cell line THP1 (Figure 2). Expression of the toxin does not induce a more cytoplasmic presence of the producing bacteria (Figure S2). Secretion of the toxin in the bacterium periplasm, host cell cytosol and supernatants are dependent on the TtsA peptidoglycan hydrolase-dependent Type 10 Secretion System (Figures 3 and 3S). The ClyA present in the supernatant from bacteria expressing an inducible ClyA lysed horse blood cells, THP1 monocytes/macrophages, but not the epithelial HeLa cells. Similarly supernatant from infected HeLa cells promotes lysis of the monocytic line THP1 (Figure 4).

The manuscript presents very interesting data. Two critical issues would need to be addressed:

1 How is ClyA secreted by the infected cells and why ClyA is not toxic to the THP1 infected cells although it has been found in the cytoplasm?

The authors should show how much toxin is present in the cytoplasm as "free toxin" or trapped in membrane structures, which would explain why infected THP1 viability is not compromised. This may be assessed by ultracentrifuging the cytosol of THP1 infected cells (1h at 100000 g) separating the soluble cytoplasm and the "membrane compartments", precipitate the proteins and perform a western blot with anti-FLAG antibody. Equal loading should be evaluated by Ponceau S staining since there is not a suitable marker to be used as loading control for soluble and membrane fractions. In this manner, it would be possible to evaluate the ratio of ClyA present as a soluble form in the cytosol and the ClyA amount retained in the membrane compartment (which would also include SCV). If most of the toxin is retained in the vesicular compartment, this means that the extracellular delivery is not dependent on the secretion in the host cytosol.

The author further discussed whether the toxin is secreted as a soluble factor or by outer membrane vesicles (OMVs). This can be tested by ultracentrifugation of the filtered supernatant from the bacterial cultures of the wt and *ttsA* mutant strains grown in culture medium with low Mg²⁺ concentration. precipitate the protein from the supernatant (mostly free soluble protein) and

the pellet (crude OMVs preparation), followed by WB analysis to assess in which fraction the toxin is enriched.

2. Why HeLa cells are resistant? Several intestinal epithelial cells with a specific focus on primary intestinal epithelial cells and other monocytic cell lines should be tested to assess whether this is a feature of epithelial cells or just HeLa.

Other points

a. Figure 1A, it would be important to present the WB showing that growth in LB medium does not induce ClyA expression. This information could be added to Supplementary Figure S1 both for ClyA and TaiA.

b. Figure 1C does not show the percentage of ClyA-positive bacteria in infected HeLa or THP1 cells in the *phoP* and *sly* mutant as comparison (as it is done in Figures 1A and 1B).

c. Figure 1E. What are the dots present in the graph? Are those individual data points, if yes, this does not match the number of 200 cells mentioned for the counting in the text describing Figure 1C. Are 200 cells counted by default in all the immunofluorescence experiments or does that number vary, depending on the experimental procedure?

d. Figure S2, how are the data reported? This information is not present in the figure legend or the Materials and Methods section. The most correct manner would be to report them as percentage of entry, which would normalize values between different biological replicates.

e. Figure 3A/B. I guess that the lysosome treatment allows the antibody to assess the ClyA protein retained inside the bacterial cytoplasm of the *ttsA* mutant. Please specify the rationale in the text, while describing the results.

Why was the experiment performed at 8h post-infection while all the other experiments have been performed at 4h (mimicking early infection) and 24h (mimicking late infection when a higher number of bacteria would be present)? It would be more convincing to have a representative micrograph that shows cells having more than one bacterium per cell.

f. Figure S3: the control without lysozyme is missing in the figure, otherwise it is difficult to interpret the data.

g. Why change the MOI (1:100 or 1:20) depending on the experiment procedure? The rationale behind this choice should be clarified.

h. Material and Methods section, line 482 "Cells were infected three times": please clarify this sentence, not clear what it means.

i. Please check Material and Methods section, line 504: the MOI is not specified ("MOI xxx").

point by point reviewer response letter

REVIEWER COMMENTS

Reviewer #1 (Remarks to the Author):

The authors detail the expression, regulation, secretion, and activity of the cytolysin ClyA in *Salmonella* serovar Paratyphi. The manuscript is well-written and elucidates the mechanism of secretion along with cell type-specific cytolytic activity. It corroborates previously known data on ClyA in serovar Typhi.

Response: We thank the reviewer for the positive assessment of our work. As requested, we have performed further experiments that are now included in the revised manuscript.

1. To validate the model proposed in Figure 5, assessing the cell type-specific activity with additional epithelial cells and macrophages would be essential.

Response: We appreciate the reviewer's insightful suggestion. We now have incorporated LDH-release assays of ClyA on human intestinal epithelial cells, including polarized and non-polarized CaCo-2 (absorptive enterocytes) and HT-29 MTX cells (mucus-producing goblet cells). Additionally, we have introduced a different human macrophage-like monocytic cell line, U937. The findings from these additional cell lines validate our previous results, indicating that macrophages remain the sole cell type susceptible to ClyA-mediated cell lysis thus far. These new results have been integrated into Figure 4 of the revised manuscript.

2. In the Western Blot presented in Figure 1, including a loading control, such as a housekeeping gene, would be valuable to confirm the observed lower expression in the mutant backgrounds. This is particularly important to ascertain whether the reduced expression is a direct result of the mutation or if these mutants exhibit decreased growth in a low Mg²⁺ medium.

Response: New Western blot analyses have been incorporated into Figure 1A and B of the revised manuscript. Both blots now also display the expression of the housekeeping protein RecA, serving as an indicator of the comparable amounts of bacteria utilized for the analyses.

Furthermore, it remains unclear whether the low Mg²⁺ medium used to induce PhoP activation is the same as the intracellular medium TTIM described in the Materials and Methods section.

Response: To clarify any confusion, TTIM and low Mg medium refer to the same media. Throughout the revised manuscript, the specification of the medium has now been updated to "low Mg²⁺ medium" for consistency.

3. The authors assert that ClyA is 'exclusively expressed within infected host cells' (lines 261-262) and 'absent in bacteria outside of host cells' (line 294). However, it appears contradictory that ClyA overexpression in LB supernatant was utilized to assess cytolytic activity. This approach seems counterintuitive, considering the stated condition-specific expression of ClyA.

Response: In vitro bacterial culture experiments were conducted to highly overexpress ClyA, utilizing rhamnose-inducible plasmids, to generate a sufficient amount of ClyA in the bacterial supernatants. This artificial induction and accumulation of ClyA in the bacterial supernatants were necessary to perform LDH release assay screens on various target cells, as depicted in Figure 4A-F. Additionally, to demonstrate the impact of natively expressed ClyA within infected epithelial cells, both wild-type

(WT) and *clyA*-deficient mutants were used to infect HeLa epithelial cells. HeLa cell supernatants were then utilized to probe for macrophage lysis. The results illustrated that this approach is sufficient to lyse nearby macrophages upon its release, as shown in Figure 4G. To clearly emphasize the focus of the different experiments, the manuscript pertaining to the described figures has been rephrased and thoroughly revised.

4. Why use horse erythrocytes for a human pathogen?

Response: We agree with the reviewer's suggestion and have now included hemolytic assays using human erythrocytes, replacing horse erythrocytes in Figure 4A.

5. The authors should provide clarification on the distribution of ClyA in *Salmonella*. While they assert that ClyA is “exclusively found in *S. Typhi* and *S. Paratyphi A* and largely absent from non-typhoidal *Salmonella* serovars” (lines 259-260) or “entirely absent in non-typhoidal *Salmonella* serovars” (line 381), a quick Blast analysis reveals its presence in several strains of serovar Indiana, Adjame, and Rubislaw... Addressing this discrepancy is crucial for defining the distribution of ClyA in *Salmonella*.

Response: *Salmonella* serotypes are broadly classified into typhoidal and non-typhoidal categories, based on the associated clinical syndromes. The reviewer is correct in noting that a Blast(p) search reveals several ClyA-positive *Salmonella* serovars beyond the prototypical Typhi and Paratyphi A typhoidal serovars. Currently, around 2,500 *Salmonella* serovars have been identified, with only a few characterized for their clinical syndromes. In our genome screens, we concentrated on better known serovars identified through the Blast(p) search for ClyA, including Indiana, Javiana, Montevideo, Panama, Adjame, and Rubislaw. The analyses revealed that, in addition to ClyA, all of these serovars also contain genes encoding for typhoid toxin, a characteristic virulence factor of typhoidal *Salmonella* such as Typhi and Paratyphi A. However, there is a lack of studies demonstrating whether these mentioned serovars exhibit typical clinical syndromes, such as the systemic infections characteristic of typhoidal serovars. Therefore, we have revised the manuscript accordingly, incorporating these points where necessary and adjusting our assertions.

6. Why SPI-1 inducing condition was used to infected THP-1 macrophages.

Response: The reviewer is correct, and SPI-1 induction may not necessarily be needed to infect professional phagocytes. However, for experimental consistency, we utilized the same infection conditions as those used for infecting epithelial cells. The use of these SPI-1 inducing conditions certainly enhances the efficiency of *Salmonella* uptake even in professional phagocytes.

Reviewer #2 (Remarks to the Author):

The authors investigated cytolysin A (ClyA) secreted by *Salmonella enterica* serovar Paratyphi A. Similar investigations on ClyA homologs in the same context have previously been published, rendering half of the data presented in this publication not so novel. Two novel findings made are: (i) ClyA is secreted in a Type 10 secretion-dependent manner (Fig. 3), and (ii) ClyA exhibits toxicity towards THP1 cells but not the SCV membrane or epithelial cells (Fig. 4). These findings are interesting. However, the authors have not described the mechanism by which ClyA exhibits selective toxicity. Furthermore, whether these in vitro observations are relevant in vivo is lacking in this manuscript. Figure 2's findings depart from the current literature. However, the authors did not provide a mechanical explanation for the disparity; if provided, this information will be helpful in the field. Without a molecular understanding, the story is simply incomplete.

We recommend that the authors conduct additional research into the mechanism by which cytolysin A specifically targets THP-1 cells but not epithelial cells or the SCV membrane. It would also be crucial to demonstrate the same cell selectivity *in vivo*, as well as the functional consequences. Without these mechanisms, the data given in this study are preliminary.

Responds: Additional research on the cell specificity of ClyA is certainly a goal we aim to achieve in future studies. Since the underlying mechanisms are currently completely unknown, comprehensive screens of mutant libraries will be necessary to identify potential alterations in host cell membranes or binding receptors specific to macrophages. Although this aspect is presently beyond the scope of our study, it will be explored in future investigations. By expanding our examination to include both intestinal epithelial cells and additional macrophages, we have further confirmed the specificity of ClyA. We believe that our discovery of ClyA's exclusive expression within the Salmonella-containing vacuole (SCV), its secretion via a novel intracellularly adapted secretion system, and its cell specificity provide sufficient data to offer greater clarity on this cytolysin adapted to the intracellular lifestyle of typhoidal *Salmonella*. The *in vivo* relevance of ClyA *per se* has already been published using a *S. Typhimurium* mouse infection model. Unfortunately, animal models for highly human-adapted typhoidal serovars such as Paratyphi A and Typhi are currently still not available.

Figures 1 and 3 should include representative microscope images. Currently, the displayed data consists of immunofluorescence quantification. The quantification data provided in different figures does not match. For instance, Figure 1C and Figure 3B provide contradictory data; the ClyA-positive bacteria proportion should be comparable.

Responds: For Figure 1C and 3B, representative microscope images are now included in supplementary Figure 1B and 3.

Figure 1C illustrates the percentage of intracellular bacteria expressing ClyA in relation to the total amount of intracellular bacteria detected. Therefore bacteria within infected cells were additionally treated with lysozyme to allow antibody access to the bacterial cytoplasm for total ClyA detection.

On the other hand Figure 3B demonstrates the detection of ClyA depending on the presence of TtsA. Here, the TtsA-dependent secretion of ClyA from the bacteria is depicted. Therefore the bacteria were not treated with lysozyme. To enhance clarity, the y-axis has been adjusted to now display the percentage of ClyA-secreting bacteria and the description of the different experimental settings is now complemented in the revised manuscript.

The description of "2 hours post-infection" lacks clarity regarding the starting point of the counting process after the addition of bacteria. Was the counting performed immediately after adding the bacteria? Furthermore, it is crucial to present additional data that demonstrates the generation of ClyA within 2 hours after bacterial invasion.

Responds: In order to provide further clarification, the revised manuscript now includes a more detailed description of the infection process. Accordingly, the counting process begins after the bacteria have incubated with the host cells for 1 hour to allow for uptake/invasion. This initiation of counting is defined as time point 0 hours post-infection (0 hpi).

Additional experiments were conducted to demonstrate ClyA expression directly after invasion at time points 0 hpi and 1 hpi. The new data are now included in Figure 1C and Supplementary Figure 1C.

Fig. 2: The key finding of Fig. 2 is to demonstrate the discrepancy in the current literature. Is there an explanation? Are there any sequence differences?

Responds: In the gene sequences, there are no differences between *S. Typhi* and *S. Paratyphi A* that would explain the different outcomes. Regarding *TaiA*, its mutation did not impact *S. Typhi* invasion into cells; only its overexpression led to a significant difference in bacterial uptake. To further verify our results, we have now included invasion and intracellular replication assays using two additional human intestinal epithelial cell lines, Caco-2 and HT-29 MTX. Additionally, we have included a Western blot analysis to demonstrate that *ClyA* is not expressed in detectable amounts before or upon contact with host cells to be infected. We believe this clearly demonstrates that cell contact or even very early infection does not induce *ClyA* expression, which would potentially impact invasion or early replication.

Minor:

Figure 1D, bottom panel: Fig. 1C indicates that around 90% of intracellular bacteria should be *ClyA*-positive. However, among the more than 20 intracellular bacteria shown, all are *ClyA*-negative.

Responds: The corresponding figure panel, which displays microscopy images, has now been replaced in the revised manuscript.

Figure 2B: There appear to be discrepancies in the graph in the lower panel. The graph in the lower panel should display individual values.

Responds: All graphs in figure 2 and all figure graphs throughout the revised manuscript now include individual data points represented as dots within the presented columns.

The approaches in Figure 3 lack detail.

Responds: The experiments for Figure 3, including the usage of lysozyme and the presented data, have now been updated and described in more detail in the revised manuscript.

Fig. 4A. Human erythrocytes appear to be more relevant than horse erythrocytes. The experiment does not include SCV-like media. The authors do not include bacteria that contain an empty vector.

Responds: In agreement with the reviewer's suggestion, lysis experiments with human erythrocytes and an empty vector control have now been included in Figure 4A.

Reviewer #3 (Remarks to the Author):

In the manuscript entitled "Cytolysin A: An intracellularly induced and secreted cytotoxin of typhoidal *Salmonella enterica* serovar *Paratyphi*", the authors have investigated the conditions for expression, localization and secretion of the cytolysin A (*ClyA*) produced by *Salmonella Paratyphi* infected cells. Specifically:

Expression of *ClyA* is induced in the culture medium with low Mg^{2+} concentration, mimicking the SVC conditions, and in host cells in a *PhoP/Q* and *SlyA*-dependent manner (Figure 1). *ClyA* does not

have a significant impact in modulating the invasion or replication of *S. Paratyphi* in HeLa cells and differentiated macrophage cell line THP1 (Figure 2). Expression of the toxin does not induce a more cytoplasmic presence of the producing bacteria (Figure S2). Secretion of the toxin in the bacterium periplasm, host cell cytosol and supernatants are dependent on the TtsA peptidoglycan hydrolase-dependent Type 10 Secretion System (Figures 3 and 3S). The ClyA present in the supernatant from bacteria expressing an inducible ClyA lysed horse blood cells, THP1 monocytes/macrophages, but not the epithelial HeLa cells. Similarly supernatant from infected HeLa cells promotes lysis of the monocytic line THP1 (Figure 4).

The manuscript presents very interesting data. Two critical issues would need to be addressed:

1 How is ClyA secreted by the infected cells and why ClyA is not toxic to the THP1 infected cells although it has been found in the cytoplasm?

The authors should show how much toxin is present in the cytoplasm as “free toxin” or trapped in membrane structures, which would explain why infected THP1 viability is not compromised. This may be assessed by ultracentrifuging the cytosol of THP1 infected cells (1h at 100000 g) separating the soluble cytoplasm and the “membrane compartments”, precipitate the proteins and perform a western blot with anti-FLAG antibody. Equal loading should be evaluated by Ponceau S staining since there is not a suitable marker to be used as loading control for soluble and membrane fractions. In this manner, it would be possible to evaluate the ratio of ClyA present as a soluble form in the cytosol and the ClyA amount retained in the membrane compartment (which would also include SCV). If most of the toxin is retained in the vesicular compartment, this means that the extracellular delivery is not dependent on the secretion in the host cytosol.

The author further discussed whether the toxin is secreted as a soluble factor or by outer membrane vesicles (OMVs). This can be tested by ultracentrifugation of the filtered supernatant from the bacterial cultures of the wt and *ttsA* mutant strains grown in culture medium with low Mg²⁺ concentration. precipitate the protein from the supernatant (mostly free soluble protein) and the pellet (crude OMVs preparation), followed by WB analysis to assess in which fraction the toxin is enriched.

Responds: We appreciate the reviewer’s suggestions and have now incorporated new data regarding the secretion of ClyA from infected host cells. As suggested, we performed ultracentrifugation of the cytoplasm of infected THP-1 cells and analyzed the supernatant and the corresponding pellet for ClyA content via Western blot analyses. The results are now included in new Figure 4D in the revised manuscript. Additionally, ClyA secretion in low Mg²⁺ medium mimicking the SCV environment, including ultracentrifugation for the detection of ClyA in supernatant or pellet, has been performed and is now included in new Figure 4E of the revised manuscript.

2. Why HeLa cells are resistant? Several intestinal epithelial cells with a specific focus on primary intestinal epithelial cells and other monocytic cell lines should be tested to assess whether this is a feature of epithelial cells or just HeLa.

Responds: We agree with the reviewers’ suggestions and have now included additional data from cell lines such as human intestinal epithelial cells and another monocytic cell line in Figure 4 of the revised manuscript. See also responds to reviewer 1.1.

Other points

a. Figure 1A, it would be important to present the WB showing that growth in LB medium does not induce ClyA expression. This information could be added to Supplementary Figure S1 both for ClyA and TaiA.

Responds: The suggested Western blot is now included in the new Supplementary Figure 1a.

b. Figure 1C does not show the percentage of ClyA-positive bacteria in infected HeLa or THP1 cells in the *phoP* and *sly* mutant as comparison (as it is done in Figures 1A and 1B).

Responds: The suggested data of ClyA-expression over time in the *phoP* and *slyA* mutants are now included in the new Supplementary Figure 1b.

c. Figure 1E. What are the dots present in the graph? Are those individual data points, if yes, this does not match the number of 200 cells mentioned for the counting in the text describing Figure 1C. Are 200 cells counted by default in all the immunofluorescence experiments or does that number vary, depending on the experimental procedure?

Responds: The dots represent 10 randomly selected and analyzed regions of interest (ROIs). All cells within each ROI (approximately 20 cells) were analyzed for ClyA-expressing bacteria. The combined analysis of these 20 cells provided the mean value of 1 dot depicted in the graph. Each column represents the mean of 10 dots or ROIs, respectively. This cell/analysis procedure has been performed for all immunofluorescence quantifications throughout the manuscript. A more detailed description has now been included in the materials and methods section of the revised manuscript.

d. Figure S2, how are the data reported? This information is not present in the figure legend or the Materials and Methods section. The most correct manner would be to report them as percentage of entry, which would normalize values between different biological replicates.

Responds: The data of now Supplementary Figure 1D, which represents the percentage of cytosolic bacteria compared to total intracellular bacteria, has now been described in more detail in the results as well as in the material and methods section of the revised manuscript.

e. Figure 3A/B. I guess that the lysosome treatment allows the antibody to assess the ClyA protein retained inside the bacterial cytoplasm of the *ttsA* mutant. Please specify the rationale in the text, while describing the results.

Why was the experiment performed at 8h post-infection while all the other experiments have been performed at 4h (mimicking early infection) and 24h (mimicking late infection when a higher number of bacteria would be present)? It would be more convincing to have a representative micrograph that shows cells having more than one bacterium per cell.

Responds: The rationales behind the lysozyme treatments have now been described in more detail in the revised manuscript. New Figure 3A now depicts images with several intracellular bacteria that have been captured at late infection states, specifically at 24 hours post-infection, where TtsA mediated secretion is most prominent.

f. Figure S3: the control without lysozyme is missing in the figure, otherwise it is difficult to interpret the data.

Responds: The old Supplementary Figure 3 has been eliminated from the revised manuscript since its message is already conveyed differently and is therefore redundant.

g. Why change the MOI (1:100 or 1:20) depending on the experiment procedure? The rationale behind this choice should be clarified.

Responds: Higher multiplicities of infections are essential when specifically more intracellular bacteria are needed for the corresponding analysis, such as Western blot analyses of intracellular bacteria. In the section of material and methods of the revised manuscript the different MOI are now explained in more detail.

h. Material and Methods section, line 482 "Cells were infected three times": please clarify this sentence, not clear what it means.

Responds: The corresponding section has now been corrected in the revised manuscript (material and methods).

i. Please check Material and Methods section, line 504: the MOI is not specified ("MOI xxx").

Responds: The MOI has now been included.

REVIEWER COMMENTS

Reviewer #2 (Remarks to the Author):

The authors addressed the majority of the concerns raised during the initial review. One minor concern remains regarding the inclusion of extra samples in Fig. 4b-f. The following samples would improve the rigor of the results: a complementation strain for *delta*clyA, *delta*clyA [vector], WT [clyA catalytic mutant], and WT [vector]. Also, please use consistent color-coding. For instance, Fig. 4g represents WT (no plasmid) in blue, while Fig. 4a indicates the equivalent sample in gray.

Reviewer #3 (Remarks to the Author):

Two main points were identified during the first review:

1. The requirement for a more detailed characterization of the toxin secretion from the host cells to add biological novelty to the results;

2. Expand the panel of cell lines tested to support the claim of specificity.

Point 2

No further comments. The authors have included two additional epithelial cells and one monocytic cell line, reproducing the resistance of cells of epithelial origin to the ClyA-induced intoxication, supporting the conclusions.

Point 1

The authors have included a new experimental set up, where cleared cytosol extracted from THP1 infected cells underwent ultracentrifugation to isolate small membrane compartments, e.g. outer membrane vesicle, from the soluble part (ultracentrifuged supernatant). The toxin was mainly present in the ultracentrifuged pellet, as assessed by western blot analysis, despite the lower total protein amount recovered as demonstrated by the Ponceau staining. A similar result was obtained from analysing the ultracentrifuged cleared cytosol from bacteria grown in a low Mg²⁺ medium, which induces ClyA expression.

These data indicate that most of the toxin is mostly associated with membrane compartments, rather than secreted as in the soluble form in the infected cells. This would explain why the infected THP1 cells are not lysed massively in the frame of the 24h infection.

Comment: the data presented support the conclusion, however, the quality of the western blot presented in the new Figure 3 is not optimal, how many independent experiments were run?

Minor points

1. Line 115-123: the following sentence "After indicated time points, the infected cells were collected and treated with lysozyme. This treatment allowed the anti-3xFLAG antibody access to ClyA within the bacteria, enabling the detection of total ClyA. By immunofluorescence microscopy, intracellular bacteria were visualized using an anti-LPS secondary antibody, which emits a red fluorescence signal. Additionally, ClyA toxins were visualized using a secondary anti-3XFLAG antibody, resulting in green fluorescence signals. Randomly selected images were quantitatively analyzed by counting the number of ClyA-positive bacteria per cell (green fluorescence signals) against the total count of bacteria per cell (red fluorescence signals)" is rather complicated to read and needs to be simplified.

2. Line 156-161 The sentence ". The induced expression of ClyA within the SCV did not result in the escape of bacteria from the SCV into the host cytoplasm, as demonstrated by quantifying cytosolic bacteria through chloroquine treatment assays (Supplementary Figure 1d). The proportion of cytosolic bacteria compared to the total intracellular bacteria remained consistent when comparing ClyA-expressing wild-type bacteria (34% cytosolic) to ClyA-negative Δ clyA mutants (31% cytosolic)." can be simplified as :

"The induced expression of ClyA within the SCV did not result in the escape of bacteria from the SCV into the host cytoplasm, since the ratio of cytosolic bacteria compared to the total intracellular bacteria remained consistent in cell infected with the wild-type bacteria (34% cytosolic) and ClyA-negative Δ clyA mutants, 43% and 31%, respectively (Supplementary Figure 1d)."

3. 615-624: The description of the infection and definition of the infection time points should be moved together with the gentamicin protection assay description in a single section labelled as "Infection".

Reviewer #4 (Remarks to the Author):

Authors have performed a systematic study to establish ClyA induction and secretion from Paratyphi A intracellularly with the help of Type 10 SS. However, the work does not provide much information in terms of novelty and contributes little to the existing knowledge regarding implication of this toxin for the bacterial pathogenesis process. Additionally, the study lacks in vivo evidence to support the in vitro assessment of the ClyA expression and its role in systemic infection.

Specific Comments:

1. The study is written in a way that it is inclined towards understanding the systemic infection caused by the Paratyphi A, especially concerning the ClyA expression and secretion. However, invasion and expression studies in the primary immune cells or primary cells from the intestine combined with the in vivo studies will be required to increase the impact of the claims and may provide novel information regarding the pathogenesis process and the contribution of ClyA in the same. At present, all the experiments are done with cell lines.
2. Authors have used different MOIs for different assays with merely stating this is the adequate ratio. Please provide a proper justification behind using 1:100 MOI vs using 1:20, and provide references where such high MOIs were used instead of the regular 10 to 50 for the infection studies.
3. Is it possible that the low pH of SCVs induces the transcriptional upregulation of the ClyA? Also, apart from the experimental limitations, what may be the reason for the 2 to 3 percent bacteria that are not present in the SCV but express clyA? Please discuss.
4. The ClyA is well known to assist *S. typhi* in invading the epithelial cells and can even promote the colonization of deep organs in mice when expressed heterologously in *S. Typhimurium*. However, the current study suggests otherwise in the case of Paratyphi without providing any reasonable explanation. Additionally, authors have discussed the membrane-bound form of ClyA as inactive and unable to contribute to the toxicity specifically in the cell types excluding immune cells. Please explain this in detail with reference to the previous studies showing similar or contrasting results.
5. Despite being one of the target intestinal cells, invasion rates in the Caco-2 cells are significantly lower than the other cell types. Please explain.
6. It is difficult to comprehend why authors have discussed and compared the ClyA in terms of expression, control, and regulation with the Typhoid toxin of *S. Typhi*. However, it would be helpful to include the comparative analysis with ClyA of *S. Typhi* or discuss the previous findings explaining the same.
7. For the cytotoxicity analysis, ClyA was overexpressed in the WT using the plasmid which may result in elevated effects different from the physiological scenario. Instead, inducing ClyA production under low magnesium conditions (as shown for other experiments) would have been more appropriate. Please discuss.
8. The study has also shown the impact of the TaiA protein on host cell invasion without proper justification and correlation with the ClyA. Please explain and also provide the previous reports for the same, specifically concerning ClyA in *S. Typhi*.
9. Authors have not italicized the scientific names all over the manuscript in a consistent manner. For example, in line 305 of the discussion, '*Salmonella enterica*' must be italicized. Also, uniformity should be maintained while writing 'Caco-2 cell'. Authors should carefully revise the manuscript for such types of typing errors.

point by point reviewer response letter

REVIEWER COMMENTS

Reviewer #2 (Remarks to the Author):

The authors addressed the majority of the concerns raised during the initial review. One minor concern remains regarding the inclusion of extra samples in Fig. 4b-f. The following samples would improve the rigor of the results: a complementation strain for $\Delta clyA$, $\Delta clyA$ [vector], WT [*clyA* catalytic mutant], and WT [vector]. Also, please use consistent color-coding. For instance, Fig. 4g represents WT (no plasmid) in blue, while Fig. 4a indicates the equivalent sample in gray.

Responds: As suggested by the reviewer, a complementation strain for $\Delta clyA$ and an empty vector control for $\Delta clyA$ and WT strains have now been included in Figure 4b-f. Additionally, the complementation strain and empty vector control for $\Delta clyA$ are now included in Figure 4g. To maintain color consistency, the color code (WT grey) has been adjusted according to the rest of Figure 4 as suggested.

Reviewer #3 (Remarks to the Author):

Two main points were identified during the first review:

1. The requirement for a more detailed characterization of the toxin secretion from the host cells to add biological novelty to the results;
2. Expand the panel of cell lines tested to support the claim of specificity.

Point 2

No further comments. The authors have included two additional epithelial cells and one monocytic cell line, reproducing the resistance of cells of epithelial origin to the ClyA-induced intoxication, supporting the conclusions.

Point 1

The authors have included a new experimental set up, where cleared cytosol extracted from THP1 infected cells underwent ultracentrifugation to isolate small membrane compartments, e.g. outer membrane vesicle, from the soluble part (ultracentrifuged supernatant). The toxin was mainly present in the ultracentrifuged pellet, as assessed by western blot analysis, despite the lower total protein amount recovered as demonstrated by the Ponceau staining. A similar result was obtained from analysing the ultracentrifuged cleared cytosol from bacteria grown in a low Mg²⁺ medium, which induces ClyA expression.

These data indicate that most of the toxin is mostly associated with membrane compartments, rather than secreted as in the soluble form in the infected cells. This would explain why the infected THP1 cells are not lysed massively in the frame of the 24h infection.

Comment: the data presented support the conclusion, however, the quality of the western blot presented in the new Figure 3 is not optimal, how many independent experiments were run?

Responds: We have now repeated the Western blot analysis, and new images with better quality are included in Figure 3d. A total of three independent experiments have been performed.

Minor points

1. Line 115-123: the following sentence "After indicated time points, the infected cells were collected and treated with lysozyme. This treatment allowed the anti-3xFLAG antibody access to ClyA within the bacteria, enabling the detection of total ClyA. By immunofluorescence microscopy, intracellular bacteria were visualized using an anti-LPS secondary antibody, which emits a red fluorescence signal. Additionally, ClyA toxins were visualized using a secondary anti-3XFLAG antibody, resulting in green fluorescence signals. Randomly selected images were quantitatively analyzed by counting the number of ClyA-positive bacteria per cell (green fluorescence signals) against the total count of bacteria per cell (red fluorescence signals)" is rather complicated to read and needs to be simplified.

Responds: The sentence has now been simplified, as suggested by the reviewer.

2. Line 156-161 The sentence ". The induced expression of ClyA within the SCV did not result in the escape of bacteria from the SCV into the host cytoplasm, as demonstrated by quantifying cytosolic bacteria through chloroquine treatment assays (Supplementary Figure 1d). The proportion of cytosolic bacteria compared to the total intracellular bacteria remained consistent when comparing ClyA-expressing wild-type bacteria (34% cytosolic) to ClyA-negative Δ clyA mutants (31% cytosolic)." can be simplified as :

"The induced expression of ClyA within the SCV did not result in the escape of bacteria from the SCV into the host cytoplasm, since the ratio of cytosolic bacteria compared to the total intracellular bacteria remained consistent in cell infected with the wild-type bacteria (34% cytosolic) and ClyA-negative Δ clyA mutants, 43% and 31%, respectively (Supplementary Figure 1d)."

Responds: The sentence has been changed as suggested.

3. 615-624: The description of the infection and definition of the infection time points should be moved together with the gentamicin protection assay description in a single section labelled as "Infection".

Responds: As suggested, both descriptions are now moved together under the Material and Methods section titled "Host cell infection and immunostaining of intracellular bacteria."

Reviewer #4 (Remarks to the Author):

Authors have performed a systematic study to establish ClyA induction and secretion from Paratyphi A intracellularly with the help of Type 10 SS. However, the work does not provide much information in terms of novelty and contributes little to the existing knowledge regarding implication of this toxin for the bacterial pathogenesis process. Additionally, the study lacks in vivo evidence to support the in vitro assessment of the ClyA expression and its role in systemic infection.

Specific Comments:

1. The study is written in a way that it is inclined towards understanding the systemic infection caused by the Paratyphi A, especially concerning the ClyA expression and secretion. However, invasion and expression studies in the primary immune cells or primary cells from the intestine combined with the in vivo studies will be required to increase the impact of the claims and may provide novel information regarding the pathogenesis process and the contribution of ClyA in the same. At present, all the experiments are done with cell lines.

Responds:

Systemic infection caused by typhoidal *Salmonella* is undoubtedly a multifactorial process involving numerous virulence factors, including but not limited to toxins, effector proteins, and metabolic adaptations. This study does not aim to explain the systemic infection of *S. Paratyphi A* solely through the example of one virulence factor. However, as demonstrated in previous studies, including those cited by the reviewer, ClyA plays a significant role in the pathogenesis of typhoidal *Salmonella* infections.

In this study, our focus is to present important findings regarding the conditions under which ClyA is expressed, its secretion mechanisms, and its specific activities towards potential target cells. Our research provides novel and direct evidence of intravacuolar expression of ClyA, a phenomenon not previously demonstrated. Furthermore, for the first time we have elucidated the mechanisms of its secretion and its specific effects on different cell types. We believe that our findings contribute significantly to the understanding of ClyA's role in typhoidal *Salmonella* pathogenesis, offering new insights into its locally defined production, secretion and specific activity towards macrophage host cells.

Within the limitations of primary cells, we have now incorporated expression and invasion assays using isolated peripheral blood mononuclear cells (PBMCs), as suggested by the reviewer. The results are now included in Figure 1c (expression) and Figure 2a (invasion).

Primary human intestinal epithelial cells, to our knowledge, remain stable only within multicellular complexes or layers such as organoids. While infection experiments using organoids are highly interesting and exciting, they are beyond the time scope of this study and will be pursued in future research endeavors.

As of today, no suitable in vivo animal model has been developed specifically for strictly human-adapted typhoidal *Salmonella*. Existing animal infection models involve using broad-host-range *S. Typhimurium* strains engineered to express typhoidal-specific virulence factors. These models, while valuable, have limitations due to the inherent differences in intracellular behavior compared to human-adapted typhoidal *Salmonella*. As the reviewer is aware, these available models have already been utilized to study the impact of ClyA on the pathogenicity process of *Salmonella*.

2. Authors have used different MOIs for different assays with merely stating this is the adequate ratio. Please provide a proper justification behind using 1:100 MOI vs using 1:20, and provide references where such high MOIs were used instead of the regular 10 to 50 for the infection studies.

Responds: A high MOI of 100 was used solely for protein analyses via Western blots, a less sensitive method, thus necessitating an adequate number of intracellular bacteria. The same MOI has been employed in various *Salmonella* infection studies. An extended explanation and corresponding references are now included in the Material and Methods section of the revised manuscript.

3. Is it possible that the low pH of SCVs induces the transcriptional upregulation of the ClyA? Also, apart from the experimental limitations, what may be the reason for the 2 to 3 percent bacteria that are not present in the SCV but express *clyA*? Please discuss.

Responds: Previous studies have shown that low pH can induce the PhoP-PhoQ Two-Component Regulatory System. Since ClyA is regulated by this system, it can be assumed that the low pH environment of the Salmonella Containing Vacuole (SCV) can induce ClyA expression.

Indeed, a very small population of cytoplasmic bacteria showed ClyA expression. This is likely within the limitations of the experimental setting, suggesting that these bacteria may have just escaped the SCV and are therefore still positive for ClyA.

4. The ClyA is well known to assist *S. typhi* in invading the epithelial cells and can even promote the colonization of deep organs in mice when expressed heterologously in *S. Typhimurium*. However, the current study suggests otherwise in the case of Paratyphi without providing any reasonable explanation. Additionally, authors have discussed the membrane-bound form of ClyA as inactive and unable to contribute to the toxicity specifically in the cell types excluding immune cells. Please explain this in detail with reference to the previous studies showing similar or contrasting results.

Responds:

We are aware of at least two studies that provide conflicting results regarding the impact of ClyA on the host cell invasion of *S. Typhi*. Sébastien Faucher et al. showed that deletion of *clyA* resulted in increased invasion rates of HeLa cells, whereas Juan A. Fuentes et al. reported decreased invasion of *S. Typhi* *clyA* mutants into Hep-2 epithelial cells. (Sébastien P. Faucher et al., *Microbiology* (2009), 155, 477–488, PMID: 19202096, J.A. Fuentes et al., *Research in Microbiology* 159 (2008), PMID: 18434098).

In our study, we observed a tendency for the *clyA* mutant to have a lower invasion rate, although this difference was not statistically significant. However, when using isolated primary immune cells, which are known to be more sensitive and less stable, the invasion rate was lower in the *clyA* mutant. Although no ClyA could be detected at the protein level before invasion, it is likely that very low amounts of ClyA are sufficient to aid *S. Paratyphi A* in invading these cell types. These new results are now included in the revised manuscript.

To our knowledge, vesicle-mediated ClyA secretion in *S. Typhi* has only been demonstrated under in vitro bacterial growth conditions using a *clyA*-overexpressing plasmid. In these studies, bacterial membrane vesicles containing ClyA were isolated and incubated with target cells, showing that ClyA in these vesicles is more active towards cells than monomeric purified ClyA. However, a membrane-bound form of ClyA within infected host cells, as discovered in our study, has not been previously described.

At this point, we do not know whether these intracellular membrane vesicles originate from bacteria, such as outer membrane vesicles (OMVs), or from the SCV, as described for typhoid toxin. In the case of typhoid toxin, localization within membrane vesicles prevents intoxication of the toxin-producing cells. We have now included a more detailed explanation of these findings in the discussion section of the revised manuscript.

5. Despite being one of the target intestinal cells, invasion rates in the Caco-2 cells are significantly lower than the other cell types. Please explain.

Responds:

The invasion experiments were repeated six times independently, and all showed a lower invasion rate for Caco-2 cells compared to the other IEC cell line, HT-29 MTX. Despite the different invasion rates and therefore intracellular bacterial loads, no impact of ClyA or TaiA was detected in either intestinal epithelial cell line. Therefore, we assume that a generally lower invasion rate does not affect ClyA or TaiA dependency. A previous *Salmonella* infection study comparing Caco-2, HT-29, and HT-29 MTX cells showed similar differences in invasion rates. Here, Caco-2 cells were less invaded than HT-29 or HT-29 MTX cells, likely due to mucus secretion, as claimed by the authors. (Mélanie Gagnon et al., "Comparison of the Caco-2, HT-29 and the mucus-secreting HT29-MTX intestinal cell models to investigate Salmonella adhesion and invasion", J Microbiol Methods. 2013 Sep;94(3):274-9. PMID: 23835135).

6. It is difficult to comprehend why authors have discussed and compared the ClyA in terms of expression, control, and regulation with the Typhoid toxin of *S. Typhi*. However, it would be helpful to include the comparative analysis with ClyA of *S. Typhi* or discuss the previous findings explaining the same.

Responds:

In this study, we demonstrate for the first time that ClyA, like typhoid toxin, is selectively expressed within the SCV. This suggests that common environmental signals within the vacuole regulate the expression and secretion of both ClyA and typhoid toxin. Consequently, throughout the manuscript, we discuss and compare ClyA with typhoid toxin. In the revised discussion section, we focus on regulatory systems known to be active within the SCV. Additional regulatory systems shown to be involved in the regulation of ClyA expression in *S. Typhi* are described and cited in the introduction section.

7. For the cytotoxicity analysis, ClyA was overexpressed in the WT using the plasmid which may result in elevated effects different from the physiological scenario. Instead, inducing ClyA production under low magnesium conditions (as shown for other experiments) would have been more appropriate. Please discuss.

Responds:

Despite significant efforts, the low magnesium medium is unsuitable for investigating ClyA-dependent cytotoxicity analyses. After 24 hours of growth to induce and accumulate sufficient amounts of ClyA in this medium, even the cell-free supernatants from *clyA* mutants lysed many tested cells, making it impossible to detect differences compared to wild-type *S. Paratyphi*. It is possible that other *Salmonella* factors induced and released under these conditions contribute to this effect. Additionally, during growth, certain metabolites may accumulate in this particular medium, lowering the pH (which was measured at 6.5 after 24 hours) or increasing its hypertonicity, thereby enhancing cell lysis. Notably, even fresh, non-cultured low magnesium medium exhibited significantly higher cytolytic effects on various target cells compared to LB medium, thereby increasing background LDH levels.

Using LB medium with artificially induced ClyA under stable conditions, which itself is non-lytic and does not induce detectable cytolytic factors including ClyA, allowed us to specifically investigate ClyA cytotoxicity on various cell types. This approach ensured that any elevated cytolytic effects observed

were specific to ClyA treatment, ruling out nonspecific or abnormal cytolytic activity in many cell lines.

8. The study has also shown the impact of the TaiA protein on host cell invasion without proper justification and correlation with the ClyA. Please explain and also provide the previous reports for the same, specifically concerning ClyA in *S. Typhi*.

Responds:

TaiA was included in our analyses since previous studies have shown its colocalization with *clyA* on the same Salmonella Pathogenicity Island, SPI-18. It has been demonstrated that *taiA* is under the same regulon as *clyA* and is co-expressed with it. Overexpression of TaiA has been shown to enhance the invasion of HeLa epithelial cells, hence its name, Typhi-associated invasin A (Sébastien P. Faucher et al., *Microbiology*, 2009, 155, 477–488, PMID: 19202096). On the other hand, deletion of *taiA* negatively impacts the invasion rate of *S. Typhi* in THP-1 macrophages but does not affect the invasion of HeLa epithelial cells.

To investigate potential interacting effects, we used a previously untested *clyA/taiA* double mutant, to assess its impact on the invasion rate of *S. Paratyphi A*. Explanation and justifications of these analyses are included in the revised manuscript. The previous reports on ClyA and TaiA of *S. Typhi* are included in the discussion section.

9. Authors have not italicized the scientific names all over the manuscript in a consistent manner. For example, in line 305 of the discussion, '*Salmonella enterica*' must be italicized. Also, uniformity should be maintained while writing 'Caco-2 cell'. Authors should carefully revise the manuscript for such types of typing errors.

Responds:

The manuscript was reviewed and corrected for italicization and consistency in name writing, as suggested by the reviewer.

REVIEWERS' COMMENTS

Reviewer #2 (Remarks to the Author):

The authors addressed my concerns and suggestions. From its initial submission, the manuscript has significantly improved, and I believe this version is ready for publication.

Reviewer #4 (Remarks to the Author):

Authors have addressed majority of the comments of this reviewer. I have no other comments.